# A meritocratic network formation model for the rise of social media influencers

Nicolò Pagan [1,2,3], Wenjun Mei [1,4 ✉], Cheng Li [1] & Florian Dörfler[1]

Many of today's most used online social networks such as Instagram, YouTube, Twitter, or Twitch are based on User-Generated Content (UGC). Thanks to the integrated search engines, users of these platforms can discover and follow their peers based on the UGC and its quality. Here, we propose an untouched meritocratic approach for directed network formation, inspired by empirical evidence on Twitter data: actors continuously search for the best UGC provider. We theoretically and numerically analyze the network equilibria properties under different meeting probabilities: while featuring common real-world networks properties, e.g., scaling law or small-world effect, our model predicts that the expected in-degree follows a Zipf's law with respect to the quality ranking. Notably, the results are robust against the effect of recommendation systems mimicked through preferential attachment based meeting approaches. Our theoretical results are empirically validated against large data sets collected from Twitch, a fast-growing platform for online gamers.

[1] Automatic Control Laboratory, ETH Zürich, Zürich, Switzerland. [2] Social Computing Group, University of Zürich, Zürich, Switzerland. [3] Social Networks Lab, ETH Zürich, Zürich, Switzerland. [4] Department of Mechanics and Engineering Science, Peking University, Beijing, China. ✉email: mei@pku.edu.cn

Especially since the explosion of online services in the past couple of decades, the impact of social networks on our lives has become more and more multifaceted: they play central roles, e.g., in the dissemination of information[1], in the adoption of new technologies[2], in the diffusion of healthy behavior[3], in the formation and polarization of public opinion[4,5]. To advance our understanding of the phenomena that take place within these platforms, there has been a recent coming-together of multiple disciplines in the study of social networks. Much of the attention has been devoted to measuring the macroscopic properties of these networks, e.g., degree, density, or connectivity, as well as to understanding the microscopic formation mechanisms[6].

Along with the rapid research progress, nowadays online social networks are also evolving into new forms. Compared with those that flourished in the first decade of the 21st century, e.g., Facebook and LinkedIn, today's most popular platforms, such as Twitter, Instagram, or TikTok, exhibit some noticeably distinguishing features. One of the most prominent differences is that these new online social platforms are directed networks that do not require mutual consent for a friendship. As such, the lifeblood of these virtual friendships is the User-Generated Content (UGC)[7,8]: in 2020, every day 500 million tweets were sent (www.dsayce.com/social-media/tweets-day/), and >80 million Instagram pictures were posted (www.omnicoreagency.com/instagram-statistics/). Thanks to the use of hashtags and integrated search engines, these new social platforms encourage their users to explore the UGC based on their interests. Thereby, users tend to follow real-life strangers and create interest-based communities.

The statistical features of the aforementioned directed UGC-based online social platforms are not the same as of real-life social networks. Yet, these directed UGC-based online platforms largely affect our societies in terms of, e.g., public opinion polarization[9], or spreading of (mis)information[10]. Moreover, since these directed platforms increase the possibility to reach wide audiences (way beyond real-life friends), users can now rapidly gain popularity[11] through their UGC, and become the so-called new influencers[12]. This trend has deeply influenced consumers' and companies' behavior in markets[13] to the point that >70% of US businesses engaged Instagram influencers to promote their products in 2017 (www.emarketer.com).

Given the potentially profound impacts of the UGC-based online social platforms on public opinions and economic behavior, as well as the spreading potential of highly influential nodes, it is important to understand (i) how the UGC relates to the emergence of tremendously fast-growing social media influencers, and (ii) what are the properties of the resulting networks.

Intuitively, better quality UGC is more likely to attract users because of its higher emotional value[14,15]. Thus, the network formation process on these platforms depends on a fundamental ingredient, the quality of the UGC. However, except for the fitness model[16] in which users are connected with probability proportional to the individuals' fitness attributes, the large multidisciplinary interest in the study of network formation has so far privileged topological and socio-economic aspects observed in offline social networks (or in online social networks which mimic them, e.g., Facebook) and neglected the effect of the UGC. For example, Stochastic Actor-Oriented Models[17], in sociology, and strategic network formation models[18], in economics, assume actors decide their ties according to a utilitarian principle based on sociological elements, such as reciprocity or network closure[19], or topological measures, e.g., in-degree or closeness centrality[20], or a combination of them[21]. These models typically lead to networks characterized by bilateral social connections and high transitivity. However, on Instagram, only 14% of the relationships are reciprocated and the average clustering coefficient is smaller than 10% (for comparison, on Facebook reciprocity and clustering score, respectively, 100% and 30%)[22]. Among the random graph models (see the seminal work by Erdös and Rényi[23] and see refs. [24–26] for extensive surveys), the preferential attachment model, proposed by Barábasi and Albert[27], in which newborn nodes choose connections proportional to the degree, has been widely acknowledged. While this mechanism leads to the scale-free effect observed in many real-world networks[28], this rich-get-richer philosophy does not justify the rise of new Instagram celebrities, i.e., the so-called Instafamous[29], whose success is built without prior fame.

The prevalence of directed, UGC-based social networks and the absence of proper mathematical models inspire us to think about their formation processes from an untouched perspective. In this paper, we propose a simple yet predictive network formation mechanism that incorporates both the utilitarian principle and the UGC quality. We assume users have a common interest and we associate them with an attribute defining the quality of their UGC. To define a UGC-based formation process, we collected a longitudinal Twitter data set on network scientists[30]. Analyzing the temporal sequence of connections, we found evidence that the formation process on directed social networks results from the individuals' continuous search for better quality UGC, measured by the alignment with the follower's interests, i.e., homophily[31], and its goodness[32]. Based on this sociological evidence, in our model agents meet with uniform or in-degree-based probability, and strategically create their ties according to a meritocratic principle, i.e., based on the UGC quality. Depending on the application, the model can incorporate users that do not actively contribute with their UGC, e.g., viewers on YouTube.

We analytically and numerically study the proposed network formation dynamics as well as the network properties at equilibrium under different meeting probability functions. First, we found that the out-degree distribution has characteristics similar to a gamma distribution, with expectations equal to the harmonic number of the network size. Furthermore, the resulting networks feature real-world social networks' properties, e.g., small diameter and small, but not vanishing, clustering coefficient, and a significant overlap in the followers' sets as a result of the homophily that characterizes agents with similar interests. Moreover, the in-degree distribution satisfies the well-known scaling property[27], but we also discover a specific pattern: the highest quality node expects to have twice (respectively, three times) as many followers as the second (respectively, third) highest, and so on. This empirical regularity has been found in many systems[33] and goes under the name of Zipf's law[34]. Notably, this result is robust against the effect of recommendation systems (which increase the visibility of popular nodes). We emphasize that, despite being widely assumed to be ubiquitous for systems where objects grow in size[35], the principle underlying the origin of Zipf's law is an open research question (see ref. [33] for a survey), and our quality-based rule reveals an intuitive, meritocratic mechanism for it. Finally, to empirically validate our model, we collected three data sets[36] from Twitch, a popular platform for online gamers (https://www.twitch.tv/p/press-center/). The successful comparison with our theoretical predictions indicates that our model, despite its simple and parsimonious form, already captures several real-world properties.

## Results

The majority of today's online social networks offer the users the possibility to actively contribute to the platform's growth by sharing different forms of UGC, according to their interests, competencies, and willingness. Looking from a different perspective, users are also exposed to the content generated by others

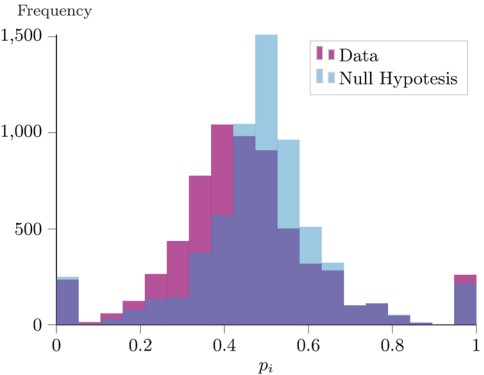

**Fig. 1 Median-rule violation on the Twitter data set.** In purple, we plot the histogram of the probability $p_i$ as defined in eq. (1). The data refer to $N = 6474$ agents out of the original 6757 by considering those with an out-degree of at least two. The median of the distribution is 0.436 (mean and std: 0.450, 0.189). In light blue, we compute the same distribution upon reshuffling the temporal sequences of the connections (null hypothesis). The median of this distribution is 0.5 (mean and std: 0.489, 0.173). The two distributions are statistically significantly different ($p$ value of Kolmogorov–Smirnov test $\ll 10^{-8}$).

on the platform. Thanks to the integrated search engines and the use of hashtags they can explore this content, discover users with similar interests, and ultimately decide to become followers. Given the limited time, they can spend on social media platforms, intuitively users seek to optimize the list of followees so as to receive high-quality content.

**Meritocratic principle**. In order to support our intuition, we collected a longitudinal Twitter data set[30] on a network composed of >6000 scientists working in the area of complex social networks. Compared with other data sets, one of the advantages is that we can expect most of the complex network scientists to be active on Twitter, given that they consistently study social networks effects. Moreover, the most popular nodes can be easily associated with renowned researchers in the field. Arguably, the number of followers can be considered as a proxy for the quality of the content generated by a user. We support this hypothesis by manually inspecting and labeling the top in-degree nodes.

Then, we enumerate the agents in decreasing in-degree order, so that agent 1 (presumably providing the best UGC) is the most followed, agent 2 is the second most, and so on. For each agent $i$, we reconstruct the temporal sequence of the outgoing connections $(i_1, i_2, \ldots, i_{d_i^{\text{out}}})$, where $i_k$ is the index (rank) of the destination of the $k$-th outgoing connection of agent $i$, and $d_i^{\text{out}}$ is her final out-degree. Then, for each agent $i$ we compute the probability

$$p_i = \frac{\left| \{ j \in \{2, \ldots, d_i^{\text{out}}\}, \text{s.t.} i_j > \text{median} \{i_1, \ldots, i_{j-1}\} \} \right|}{d_i^{\text{out}} - 1}, \quad (1)$$

that estimates the likelihood that a new connection of agent $i$ is higher (in ranking) than the median of the previous connections. At one extreme, for $p_i = 0$ the sequence $(i_1, i_2, \ldots, i_{d_i^{\text{out}}})$ is such that every new followee has a rank smaller than (or equal to) the median of the current list of followees' ranks, i.e., $i_j \leq \text{median} \{i_1, \ldots, i_{j-1}\}$. As such, the rolling median, i.e., the median computed among the first $k$ elements, is always non-increasing in $k$, and the user is continuously seeking better UGC. Conversely, for $p_i = 1$ the rolling median is always increasing.

In Fig. 1 we compare the histogram of the empirical distribution of $p_i$ (in purple) with the null hypothesis, plotted

in blue, in which we remove the temporal-ordered pattern of the sequence. The elements in agent $i$'s followees set $\mathcal{F}_i^{\text{out}} := \{i_1, i_2, \ldots, i_{d_i^{\text{out}}}\}$ are randomly re-ordered: in the new sequence $(\bar{i}_1, \bar{i}_2, \ldots, \bar{i}_{d_i^{\text{out}}})$, the $k$-th element of the list is drawn uniformly at random from $\mathcal{F}_i^{\text{out}} \setminus (\bar{i}_1, \ldots, \bar{i}_{k-1})$. The null hypothesis has a median value of ~0.5, which is easy to interpret: if the sequence is completely random, adding an extra element to the partial sequence has a 50% probability of being above the median, and 50% probability of being below it. Comparing the two distributions, we notice that empirical data tend to have a decreasing quality-ranking sequence of followees. As the difference is statistically significant, we can reject the hypothesis that the temporal sequence is random.

Ultimately, this empirical evidence confirms our intuition, i.e., users tend to continuously increase the quality threshold of the new followees. This characteristic, though, is missing in the network formation literature, as there is typically no quality associated with the users. For instance, in the preferential attachment model[27], each user selects $m$ followees proportional to their in-degree. If the network is large enough, the probability of selecting a node $k$ in the $d$-th draw does not depend on $d$. Therefore, the temporal sequence of connections of the preferential attachment model is similar to the null hypothesis. Moreover, even in the fitness model[16] where the quality (fitness) is considered, users tend to connect to high-quality nodes first, rather than later.

**Quality-based model**. To formalize our quality-based model, we consider the unweighted directed network among $N \geq 2$ agents whose UGC revolves around a specific common interest, e.g., a particular traveling destination. We denote the directed tie from $i$ to $j$ with $a_{ij} \in \{0, 1\}$, where $a_{ij} = 1$ means $i$ follows $j$. Then, we assume there are no self-loops and that each agent $i$ can only control her followees $a_{ij}$ but not her followers $a_{ji}$. Similarly to the approach in the fitness model[16], we endow each actor $i$ with an attribute $q_i$, drawn from a probability distribution, e.g., uniform, normal, exponential distribution, that describes the average quality of $i$'s content[32], e.g., a picture taken at that traveling destination. As will be manifested later, our model predictions are independent of the numerical representation of these qualities, which could be somehow subjective and arbitrary. Instead, in our model, only the ordering of the individual qualities matters. Therefore, the choice of the underlying probability distribution does not affect any of the following results, contrary to the fitness model[16].

The quality $q_i$ can be seen as the expectation of a Bernoulli random variable $Q_i$ describing the probability of followers liking agent $i$'s content. Higher values of $q_i$ are then associated with better UGC. A value of zero, instead, can be used to model users that do not produce any UGC. With this setup, the model can be directly applied to the platforms, e.g., YouTube or Twitch, in which users can be partitioned into two classes, i.e., the content creators and their followers (or viewers, see Supplementary Note 1).

We then consider a sequential dynamical process starting from the empty network, where at each time-step $t \in \{1, 2, \ldots\}$ each actor $i$ picks another distinct actor $j$ chosen randomly from a probability distribution on $\{1, \ldots, i-1, i+1, \ldots, N\}$. In the following theoretical analysis, we consider the uniform distribution. However, we also integrate into our discussion the numerical comparison between uniform distribution and the in-degree-based preferential attachment meeting process.

To reflect the meritocratic principle, we base the tie formation decision on the comparison between $i$'s current followees' and $j$'s

qualities. Let the payoff function of agent $i$ measure the maximum quality received by $i$, i.e.,

$$V_i(t) := \max_{j \in \mathcal{F}_i^{\text{out}}(t)} q_j, \qquad (2)$$

where $\mathcal{F}_i^{\text{out}}(t) := \{j, \text{ s.t. } a_{ij}(t) = 1\}$ denotes the set of $i$'s followees at time $t$. According to a utility maximization principle, we define the update process through the following rule:

$$a_{ij}(t+1) = \begin{cases} 1, & \text{if } q_j > V_i(t), \\ a_{ij}(t), & \text{otherwise}, \end{cases} \qquad (3)$$

meaning that $i$ will add $j$ in her followees' set if $j$ provides better quality content compared to $i$'s current followees. Note that, if $i$ finds a node that already belongs to her set of followees, the connection will not be re-discussed. While, intuitively, this may lead to a large out-degree, we will show that this is not the case because the cost of good-quality connections is infinitely low, but the cost of poor-quality ones is infinitely high.

We emphasize that the choice of the payoff function eq. (2) reflects the natural continuous chase for the maximum quality[37] (exploitation) while minimizing the effort owing to non-improving connections. Alternatively, one could consider smoother payoffs such as the followees' average quality

$$V_i(t) := \frac{\sum_{j \in \mathcal{F}_i^{\text{out}}(t)} q_j}{|\mathcal{F}_i^{\text{out}}(t)|},$$

as in ref. [32], which allows for more exploration, at the expense of the least-effort principle. Both definitions, though, share the same meritocratic principle previously discussed.

**Convergence**. A natural question that arises when defining a dynamical process is whether it reaches or not an equilibrium. In what follows we show that an equilibrium state is reached almost surely. To do so, from now on we assume that there exist no two agents with equal quality. Yet, we emphasize that the model and our analysis can be generalized to the case where two or more agents have the same quality (see Supplementary Fig. 6).

Then, without loss of generality, we can re-order the agents by decreasing quality $q_i$, i.e.,

$$q_1 > q_2 > \ldots > q_N.$$

In this way, agent 1 is the top-quality agent, agent 2 is the second-best, and so on. According to our dynamics, any node $i > 1$ creates new links towards increasingly quality agents, until finding the top-quality node 1. Likewise, node 1 creates new links until finding the second-highest quality agent, node 2. Convergence is guaranteed by the following theorem, proven in the Methods section.

**Theorem 1.** (Convergence) For any set of qualities $\{q_1, \ldots, q_N\}$, the network reaches an equilibrium almost surely. Moreover, the probability of reaching equilibrium within $t$ time-steps reads as follows:

$$\begin{aligned} &P[\text{An equilibrium is reached within } t \text{ time steps}] \\ &= P[a_{12}(t) = 1, a_{21}(t) = 1, \ldots, a_{N1}(t) = 1] \\ &= P[a_{12}(t) = 1] \times P[a_{21}(t) = 1] \times \ldots \times P[a_{N1}(t) = 1] \\ &= \left(1 - \left(\frac{N-2}{N-1}\right)^t\right)^N. \end{aligned}$$

From the above theoretical cumulative distribution function, we derive the expected number of time-steps to reach equilibrium. The result, shown in Fig. 2 (in blue), indicates that the number of time-steps grows almost exponentially in the network size $N$.

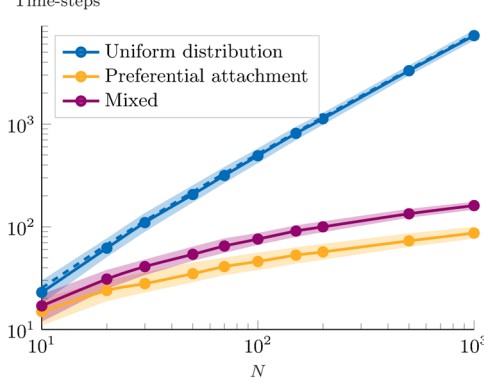

**Fig. 2 Numerical analysis of the number of time-steps until convergence.** For each value of $N$, we run 1000 simulations, in three different settings for the meeting process probability distribution: in blue, we use a uniform distribution (the dashed line also shows the theoretical results), in orange, we use a preferential attachment mechanism, and, in purple, we use a mixed distribution (50% chance of uniform distribution and 50% chance of preferential attachment, for each meeting). The data points indicate the median value. The shaded area indicates the first and third quantile.

For comparison, we consider two alternative scenarios for the meeting probability distribution. In orange, we use a preferential attachment approach: at each time-step, each agent $i$ meets a distinct agent $j$ according to a probability that is proportional to the current in-degree of the agent $j$. By doing so, the quality-based dynamics (in the follow/not follow decision) are combined with a meeting process that coarsely resembles the effect of recommendation systems (increasing the visibility of high in-degree nodes). In purple, the meeting agent is chosen with 50% probability, from the uniform distribution, and with the remaining 50% probability, according to the preferential attachment mechanism. As the numerical results indicate, the preferential attachment-based meeting process significantly reduces the number of required time-steps, even when mixed with a uniform distribution.

**In-degree distribution**. The network structure at equilibrium is a consequence of the probability distribution ruling the meeting process, and of the meeting order itself. Even though there can be multiple equilibria, some network macroscopic properties can be statistically described. A preliminary observation is that, at equilibrium, every other node follows node 1, and node 1 follows node 2. Before studying the stationary state, we statistically describe the transient in-degree distribution as a function of the quality ranking $i$. In particular, we provide an analytical formula for the expected in-degree of each node as a function of $i$, as stated in the following theorem.

**Theorem 2.** (In-degree distribution) The probability that node $i$ is followed by node $j \neq i$ after $t > 0$ time-steps is:

$$P[a_{ji}(t) = 1] = \begin{cases} \bar{p}_i(t) := \frac{1}{i-1}\left(1 - \left(\frac{N-i}{N-1}\right)^t\right), & \text{if } j < i, \\ \underline{p}_i(t) := \frac{1}{i}\left(1 - \left(\frac{N-i-1}{N-1}\right)^t\right), & \text{if } j > i. \end{cases} \qquad (4)$$

Moreover, the probability of node $i$ having in-degree $d_i^{\text{in}}(t) = d \in [0, N-1]$ after $t$ time-steps is given by

$$\begin{aligned} &P[d_i^{\text{in}}(t) = d] \\ &= \sum_{k=0}^{d} \binom{i-1}{k} \bar{p}_i^k (1 - \bar{p}_i)^{i-1-k} \binom{N-i}{d-k} \underline{p}_i^{d-k} (1 - \underline{p}_i)^{N-i-(d-k)}, \end{aligned} \qquad (5)$$

**a**

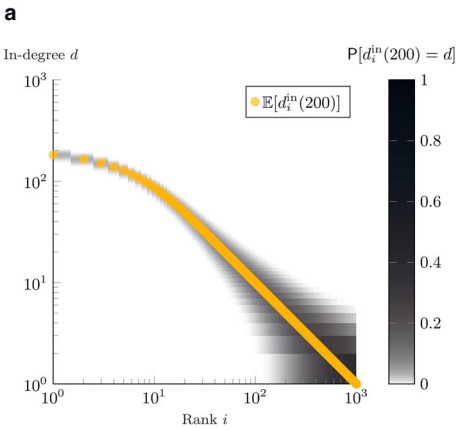

**b**

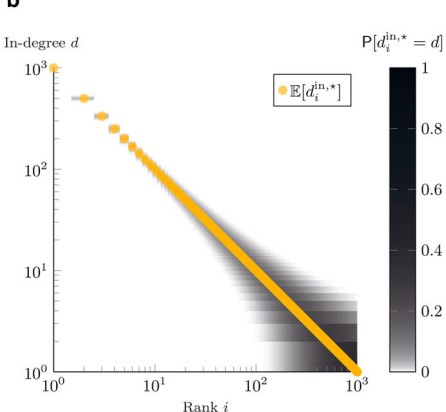

**Fig. 3 In-degree, theoretical analysis.** Given a network of $N = 1000$ agents, the plots show the probability density functions of the in-degree (derived through eq. (5) from eq. (4) and eq. (7)) as a function of the rank $i$. In orange, we also plot their expected value (equations eq. (6) and eq. (8)). The plot in **a** refers to the theoretical results at $t = 200$, in **b** to the equilibrium. The plots are shown in log–log scale, to emphasize Zipf's law: $\mathbb{E}[d_i^{in,\star}] = N/i$.

where we omitted the time-step dependency on $\bar{p}_i$ and $\underline{p}_i$. Finally, the expected in-degree of node $i$ after $t$ time-steps reads as:

$$\mathbb{E}[d_i^{in}(t)] = \frac{N}{i} - \left( \left( \frac{N-i}{N-1} \right)^t + \frac{N-i}{i} \left( \frac{N-i-1}{N-1} \right)^t \right). \quad (6)$$

The proof of the theorem is provided in Supplementary Note 1, whereas in Fig. 3a we show the probability density function, together with its expectation, for a network of 1000 agents, after $t = 200$ time-steps (non-stationary state). The following corollary, instead, studies the probability density functions upon reaching the network dynamics equilibrium.

**Corollary 1**. At equilibrium, the probability that node $i$ is followed by node $j \neq i$ is:

$$\mathsf{P}[a_{ji} = 1] := \lim_{t \to \infty} \mathsf{P}[a_{ji}(t) = 1] = \begin{cases} \bar{p}_i^\star := \frac{1}{i-1}, & \text{if } j < i, \\ \underline{p}_i^\star := \frac{1}{i}, & \text{if } j > i, \end{cases} \quad (7)$$

and the expected in-degree of node $i$ reads as:

$$\mathbb{E}[d_i^{in,\star}] = \begin{cases} N-1, & \text{if } i = 1, \\ \frac{N}{i}, & \text{otherwise.} \end{cases} \quad (8)$$

**Proof**. First, we derive the probability of node $i$ being followed by node $j$ at equilibrium by taking the limit $t \to \infty$ in eq. (4):

$$\lim_{t \to \infty} \mathsf{P}[a_{ji}(t) = 1] = \begin{cases} \bar{p}_i^\star := \frac{1}{i-1}, & \text{if } j < i, \\ \underline{p}_i^\star := \frac{1}{i}, & \text{if } j > i. \end{cases}$$

Similarly, taking the limit for $t \to \infty$ of eq. (6) yields exactly to

$$\mathbb{E}[d_i^{in,\star}] = \begin{cases} N-1, & \text{if } i = 1, \\ \frac{N}{i}, & \text{otherwise}. \end{cases}$$

According to the above result, at equilibrium the best content provider, node 1, receives $N-1$ connections, node 2 has $N/2$ expected followers, node 3 has $N/3$, and so on. The result can be intuitively reached with the following plausible reasoning: any user that has not yet found node 1 nor node 2, has the same probability of finding any of the two in the coming time-step. In expectation, in half of the cases, the user will become a follower of node 2 before finding and following (necessarily) node 1. In the other half of the case, she will find node 1 before having seen node 2. Thus, the expected number of followers of node 2 is half of the expected number of followers of node 1. The reasoning can be extended for any node $j > 2$.

Such a regular scaling property is called Zipf's law[34] and it is illustrated in Fig. 3b, where we plot the expected in-degree of each node as a function of its quality ranking, together with the probability density functions. In the log–log scale, the expected in-degree perfectly matches a line with coefficient $-1$. Real-world evidence of Zipf's law has been reported in many systems, including firm sizes[38], city sizes[33], connections between web-pages[39].

To empirically validate our results in the context of online social networks, we collected and analyzed data from Twitch, an online social media platform focusing on video streaming, including broadcasts of gameplay, e-sports competitions, and real-life content. Over the past decade, Twitch gradually became one of the most popular social media platforms, reaching up to four million unique creators streaming each month and 17 million average daily visitors (https://www.twitch.tv/p/press-center/), and serving as a virtual third place for online entertainment and social interactions[40]. Our data sets consist of the followership networks in three different categories, i.e., poker, chess, and art, where users can live-watch broadcasters playing or discussing these arguments. Among them, we only retain the first two categories, as the third one did not satisfy the criterion of having a baseline community of interested users. The description of our crawling method and of the collected data are provided in the Methods section and in Supplementary Note 3. In Fig. 4, we show the in-degree of the 15 most followed users in chess and poker as a function of the rank. The empirical data are fitted via linear regression (in the log–log plot) and look strikingly similar to Zipf's law.

**Zipf's law is more than a power-law**. The peculiarity and apparent ubiquity of Zipf's law have triggered numerous efforts to explain its origins[33]. Despite being a discrete distribution, Zipf's law is often associated with the continuous Pareto distribution, better known as power-law[41]. For this reason, it is frequently seen as the result of a linear preferential attachment process[27] based on Gibrat's rule of proportional growth[42] which leads to Yule-Simon (power-law) distributions[43]. However, as noticed in[35], there is more than a power-law in Zipf: although a power-law distribution is certainly necessary to reproduce the asymptotic behavior of Zipf's law at large values of rank $i$, any random sampling of data does not lead to Zipf's law and the deviations are dramatic for the largest objects. In particular, Zipf's law emphasizes the relation among the top-ranking elements, which essentially correspond to the most important nodes, i.e., the network influencers. The typical Zipf's sequence $N, N/2, N/3, \ldots$ can be observed only if data constitutes a coherent set[35]. Thus, Zipf's law is more insightful than just a power-law.

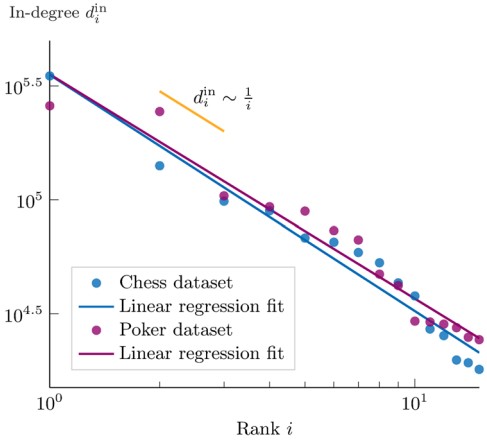

**Fig. 4 Zipf's law on empirical Twitch data.** The log–log plot shows empirical evidence of Zipf's law in our Twitch data sets. In blue, we report the data on the number of followers of the top 15 users (broadcasters) in the chess category, as well as their linear regression fit of coefficient −1.04 (Pearson coefficient, −0.98, $R^2 = 0.96$, $RMSE = 0.16$). In purple, the poker data set shows a similar fitting coefficient of −0.98 (Pearson coefficient: −0.97, $R^2 = 0.95$, $RMSE = 0.17$). Both fits are very close to the theoretically expected Zipf's law (of coefficient −1) shown in orange.

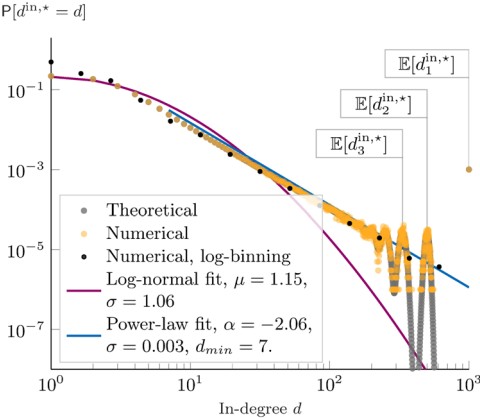

**Fig. 5 In-degree probability density function.** In gray, the theoretical result for $N = 1000$ agents is shown. In orange, we plot the numerical distribution resulting from 1000 simulations upon reaching equilibrium. In black, the numerical distribution is shown after using the standard logarithmic binning of data. Finally, we fit the numerical data with a power-law using the algorithm in Clauset et al.[44] (blue) and with a log-normal distribution (purple).

The difference with a Pareto distribution becomes evident when considering our in-degree probability density function, which can be analytically derived by using the result of the previous theorem and computing the average of each user's probability density function

$$P[d^{in,\star} = d] = \frac{1}{N}\sum_{i=1}^{N} P[d_i^{in,\star} = d].$$

As shown in Fig. 5, the theoretical in-degree probability density function follows a power-law of coefficient $\hat{\alpha} = -2.06 \pm 0.003$ ($p$ value $< 10^{-8}$), which is not surprising since the expected in-degree is distributed according to a Zipf's law (see again the discussion in ref. [41]). However, notice that the power-law fit does not hold in the region of low-in-degree nodes ($d_{min} = 7$,

according to the Clauset algorithm[44]). In this region, the log-normal distribution is a better fit compared to the power-law[45]. On the other hand, the fit with a log-normal distribution is worse (compared to the power-law) in the heavy tail of our distribution, i.e., in the region of high-quality nodes. However, the most noticeable difference is that, contrary to both the power-law and the log-normal curve, our theoretical distribution is not monotonically decreasing, especially in its right tail. The probability of having a node of in-degree $N/2$ is slightly larger than the probability of having a node of in-degree $N/2 \pm \alpha$, for some range of $\alpha \geq 1$. Moreover, the probability of having a node of in-degree in a neighborhood of $N/2$ is larger compared with the power-law fit (chosen as baseline). Conversely, our theoretical probability of having a node of in-degree slightly smaller than $N-1$ is negligible ($<10^{-8}$), stating that the second-best node cannot be too close to the best node. This is in sharp contrast with the monotonic Pareto and log-normal distributions.

In synthesis, our probability distribution differs from Pareto and log-normal distributions when focusing on the network influencers. In this range, i.e., in the extreme right tail, our theoretical distribution is similar to its power-law fit, only after logarithmically (and thus coarsely) binning the data (as common practice when empirical data are scattered and the distributions are continuous). Yet, without such arguably coarse binning of the data, our theoretical distribution can predict more accurately the Zipf's regularity found on the top influencers of real-world networks, as in the Twitch data set shown in Fig. 4. We finally refer to Supplementary Note 4 (see in particular, Supplementary Figs. 12, 13) for a detailed analysis of the empirical in-degree distribution.

**Preferential attachment meeting process.** The theoretical results on the nodes' in-degree distributions are derived based on a uniform distribution meeting process. In this scenario, every user has the same probability of being found. However, most social media platforms personalize the content users are exposed to, thus it is fundamental to understand what can be the impact if some users have more visibility than others. In order to do so, we numerically studied the results of a preferential attachment in-degree-based meeting process, which mimics the idea that popular users might get promoted (in terms of visibility) by the platform's recommendation system. Compared with the preferential attachment model[27], though, here we only alter the probability of being found, but the connection will still depend on the meritocratic principle of eq. (3). In Fig. 6, we report the results of a mixed scenario in which the potential followee is chosen with 50% probability from the uniform distribution, and with the remaining 50% probability from an in-degree based distribution. We refer to Supplementary Note 2 (see in particular Supplementary Fig. 1) for more details on the numerical comparison between uniform distribution, mixed preferential attachment, and pure preferential attachment. Compared to the uniform distribution scenario in Figs. 3 and 5, the introduction of a mixed preferential attachment-based meeting process slightly increases the variance of the in-degree probability distribution of each agent. In this scenario, it becomes possible that some low-quality agents get an initial (i.e., in the early stage of the network formation process) advantage (purely by chance), which gets reinforced by the (mixed) preferential attachment mechanism. Yet, this effect quickly fades away, because potential followers still undergo the quality threshold rule of eq. (3). On the other hand, the (mixed) preferential attachment may penalize some other agents, which receive fewer followers than what they would expect with the uniform distribution process. It is remarkable that, even after introducing the mixed preferential attachment process, the correlation between quality and followers persists (on average): the higher the quality, the higher the average number of followers. Even more importantly, Zipf's relation is robust under mixed preferential

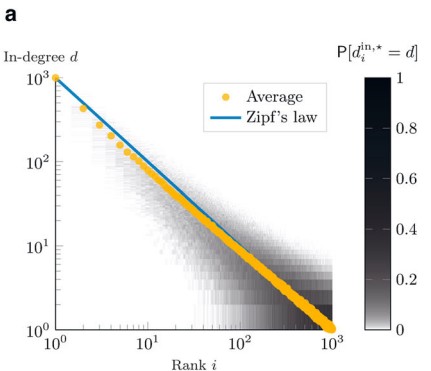
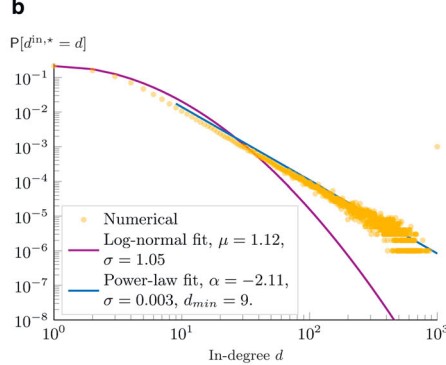

**Fig. 6 Numerical analysis of the preferential attachment effect on the in-degree distribution.** Numerical results of 1000 simulations with 1000 nodes for a mixed process (with 50% probability, the potential followee is chosen from a uniform distribution, and with the remaining 50% from a preferential attachment mechanism). **a** The color map shows the in-degree probability density function, as a function of the quality rank. In orange, the average in-degree. In blue, Zipf's law equals the expected in-degree in the uniform distribution scenario. **b** The numerical probability density functions (orange) and their power-laws (blue) and log-normal (purple) fits are shown.

attachment-based meeting processes (e.g., recommendation systems). Similarly, on the right column of Fig. 6, the average in-degree probability distribution function is strikingly similar to the one obtained with the uniform distribution scenario.

**Out-degree distribution.** Similarly to the in-degree, we can study the statistical distribution of the nodes' out-degree at equilibrium. First, notice that, contrary to the in-degree, the out-degree distribution does not depend on the rank, but is uniform for all the nodes in the network. In fact, according to the dynamics, each node creates connections to increasing quality nodes, until reaching its own equilibrium. Let $d_N^{out,\star}$ be the random variable describing the out-degree (at equilibrium) of a general node $i$ in a network of $N$ agents. First, note that the probability of node $i > 1$ finding node 1 (or node 1 finding node 2) at her first choice is simply:

$$P\left[d_N^{out,\star} = 1\right] = \frac{1}{N-1}.$$

Conversely, the probability of having maximum out-degree corresponds to the probability of meeting all the other nodes in increasing quality ordering, i.e.,

$$P\left[d_N^{out,\star} = N-1\right] = \frac{1}{(N-1)!}.$$

Obviously, a node cannot have an out-degree larger than $N-1$ in a network of $N$ agents, thus $P\left[d_N^{out,\star} = d\right] = 0$, for all $d \geq N$. Similarly, $P\left[d_N^{out,\star} = d\right] = 0$, for all $d < 1$, every node must have out-degree of at least 1. Then, we denote as $C_1$ the random variable describing the first meeting of $i$. $C_1$ can take any value $j \in \{1, \ldots, i-1, i+1, N\}$, with uniform probability $1/(N-1)$. Then, the probability $P\left[d_N^{out,\star} = d\right]$, for $N \geq 2$ and $1 \leq d \leq N-1$, can be described recursively as follows:

$$P\left[d_N^{out,\star} = d\right] = \sum_{j=1}^{i-1} P[C_1 = j]P\left[d_j^{out,\star} = d-1\right]$$
$$+ \sum_{j=i+1}^{N} P[C_1 = j]P\left[d_{j-1}^{out,\star} = d-1\right] \quad (9)$$
$$= \frac{1}{N-1} \sum_{j=1}^{N-1} P\left[d_j^{out,\star} = d-1\right].$$

In other words, $P\left[d_N^{out,\star} = d\right]$ is equal to the sum of the probability of agent $i$ choosing $j \in \{1, \ldots, i-1, i+1, \ldots, N\}$ as first choice times the probability of having out-degree $d-1$ in a network with $j$ remaining nodes (the node itself and the $j-1$ nodes to which she can still connect to).

Thanks to eq. (9), we can describe the out-degree probability functions by means of recursion on the network size $N$. The result for different values of $N$ is pictured in Fig. 7a. The out-degree distribution exhibits non-monotonic properties (first increasing, then decreasing) and quickly vanishing tail: extremely large values are particularly rare. Even though the theoretical out-degree distribution cannot be directly associated with any known distribution, it is very close to a gamma distribution (or to a Poisson distribution), as shown in Supplementary Fig. 2 (see also the discussion in Supplementary Note 2). Thanks to its non-monotonic nature and its fast decay, there are also some similarities with the log-normal distributions (whose decrease is more than linear in the log–log plot). On the other hand, it differs more significantly from a power-law distribution. Despite being observed in the in-degree distributions of many real-world social networks, power-laws do not always properly describe their out-degree distributions. Empirical evidence on the asymmetry between the in- and out-degree distributions is found on Twitter[46] and YouTube[47]. In particular, several empirical studies on the out-degree distributions of online social networks highlight a much faster decrease in the probability of seeing very high out-degree nodes[48–50]. Albeit some influencers on YouTube, Instagram, or Twitch, might have one million followers, it is hard to imagine that some users follow several thousand or even a million of other users, since every single action of the following someone involves a decision-making process and at least a click on the Follow button. As a result, the out-degree distributions of these platforms feature a clear cut-off (sometimes even artificially imposed to prevent fake-user accounts) on the order of roughly a thousand connections (for network samples, this clearly reduces even further), instead of a heavy tail as featured by the power-law.

In our Twitch data sets, we find evidence of this fast drop in the frequency of the high out-degree nodes. As shown in Fig. 8, the out-degree probability density function is concentrated in the range $d \in [1, 10]$, with the 99-percentile being at $d = 15$ ($d = 19$) in the chess (poker) data set. As a comparison, our theoretical distribution would predict the 99-percentile to be at $d \sim 10$, while a power-law of classical exponent $-2$ predicts it at $d = 100$. The maximum out-degrees are found to be, respectively, $d = 151$ and $d = 142$, which are higher than the maximum out-degree of the theoretical distribution $d \sim 20$, but at least three orders of magnitude smaller than the maximum predicted by a power-law of exponent $-2$, i.e., from the maximum in-degree of the empirical distribution. At first glance, the theoretical and empirical results in Figs. 7a and 8 show some differences. In particular, compared with the data the empirical distribution exhibits a larger frequency of low out-degree nodes. The

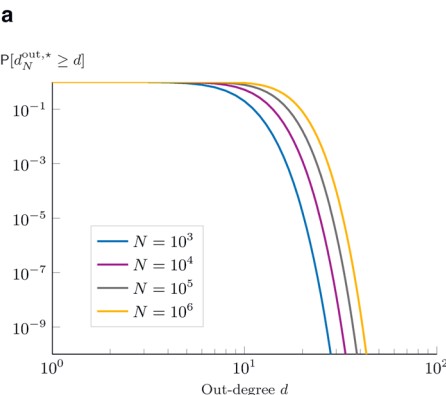
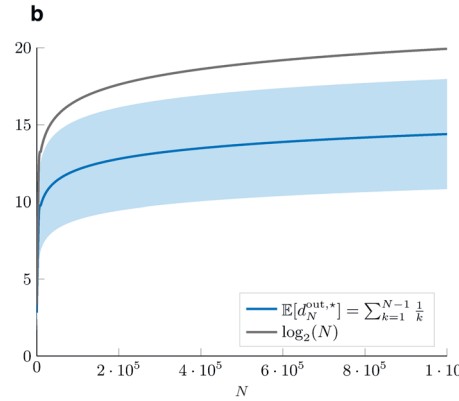

**Fig. 7 Theoretical analysis of the out-degree distribution. a** Shows the out-degree complementary cumulative distribution function for different size $N$ of the network. Note that the support of the distributions is $[1, N{-}1]$ but with very quickly vanishing right tails. **b** Shows, in bold blue, the expected out-degree as a function of the number of agents $N$ in the network. In gray, the function $\log_2(N)$ shows a similar growth. The shaded area represents the confidence interval obtained with one standard deviation computed from the out-degree distribution using the definition, $Var[d_N^{out,\star}] = \sum_{d=0}^{N-1} (d - \mathbb{E}[d_N^{out,\star}])^2 \, \mathsf{P}[d_N^{out,\star} = d]$, and the recursive formula eq. (9).

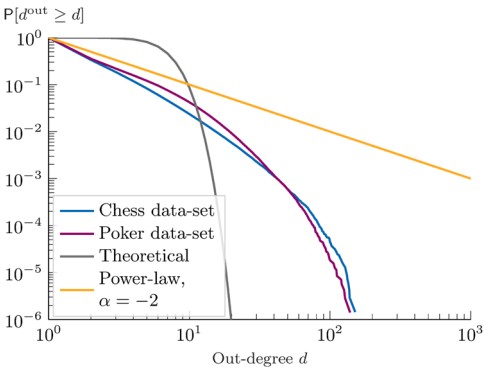

**Fig. 8 Empirical analysis of the out-degree distribution.** The plot shows the complementary cumulative distribution function of the out-degree of the Twitch data sets concerning the users in the categories chess (blue) and poker (purple). For comparison, we plot the results of our theoretical distribution with $N = 350$ (gray), and of a power-law of exponent $-2$ (orange).

difference can be due to the fact that the network is still in the formation process, and many users only joined it recently. Therefore these most recent users have just started their search and there is an abundance of low out-degree nodes. We conjecture that another possible reason may be the recommendation systems behind the Twitch platform (we explored this effect by studying the numerical out-degree distribution under preferential attachment-based meeting process, see Supplementary Fig. 3). To better understand this inconsistency, in Supplementary Fig. 15, we studied the stacked frequencies of the out-degree of the followers of the 15 most followed nodes in each data set.

From the recursive description of the probability density function in eq. (9), it is possible to compute the expected nodes' out-degree, as stated in the following theorem.

**Theorem 3**. (Out-degree distribution) At equilibrium, the nodes' expected out-degree $\mathbb{E}[d_N^{out,\star}]$ in a network of $N \geq 2$ agents equals the $(N-1)$-th harmonic number:

$$\mathbb{E}[d_N^{out,\star}] = \sum_{k=1}^{N-1} \frac{1}{k}.$$

According to the theorem, the expected out-degrees as a function of $N$ are given by the harmonic sequence: 1, 3/2, 11/6, 25/12, ….

The proof provided in Supplementary Note 1 is built on the derivation of the out-degree probability distribution, however, the result can be intuitively derived from the following observations: the expected out-degree is uniform across all the nodes, and the sum of the expected out-degree equals the sum of the expected in-degree of all the nodes. From the result on the in-degree distribution (see eq. (8)), the sum of the expected in-degree is equal to $(N-1) + N/2 + N/3 + … + 1$. Thus, the expected out-degree is equal to $(N-1)/N + N/(2N) + N/(3N) + … + 1/N$, which in fact corresponds to the $N-1$-th harmonic number. As shown in Fig. 7b, the expected out-degree has a similar growth compared with the base-2 logarithm of the network size $N$.

**Diameter and clustering**. Since each node is connected, on average, to roughly $\log_2(N)$ other nodes, intuitively, the network should feature the small-world property. In Fig. 9a we provide some numerical results on the average network diameter and average nodes' distance for different network sizes. According to the results, also the network diameter has a growth rate similar to the base-2 logarithm of the network size. Moreover, the average node's distance has even slower growth, with a value of 5 for networks of $10^4$ nodes. These results match the empirical observations on the small-world property of real-world networks, according to which the distance between two randomly chosen nodes grows proportionally to the logarithm of network size[51]. Moreover, they are quantitatively similar to previous findings on, e.g., an Instagram network sample[52] (of size equal to 44 thousand nodes) where the diameter is found to be 11, and the average distance 3.16.

As such, our theoretical model already captures some of the most widely observed features of real-world networks, i.e., the in-degree scaling and the small-world properties. A third common feature is the high clustering coefficient. Even though it has been shown that the average value on directed social networks such as Instagram or Twitter, is smaller than on other undirected social networks, e.g., Facebook, it remains on the order of 10%[22]. To compute the clustering coefficient for directed networks, we adopted the class out in the taxonomy of[53] in which a triad around node $i$ is closed when $i$ follows two distinct agents $j$ and $k$, and there exists a tie from $j$ to $k$ or from $k$ to $j$. Accordingly, the clustering coefficient of $i$ reads as

$$c_i = \frac{\sum_{j \neq i} \sum_{k \neq i,j} a_{ij} a_{ik} a_{jk}}{\sum_{j \neq i} \sum_{k \neq i,j} a_{ij} a_{ik}}.$$

The numerical results pictured in Fig. 9b indicate that the average clustering coefficient is monotonically decreasing in the

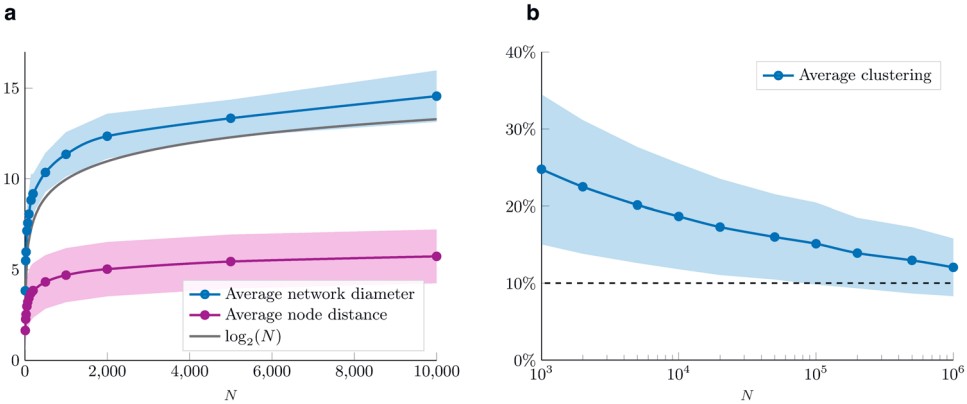

**Fig. 9 Diameter and clustering analysis.** Numerical analysis of **a** the average diameter and average nodes' distance, and **b** average clustering coefficient for different network sizes. The shaded area indicates one standard deviation from the average value. For the plot on the left, we run 100 simulations for each value of $N$ (except for the last data-point for which we run 25 simulations). For the plot on the right, we run 100 simulations if $N \leq 10^4$, and one simulation otherwise.

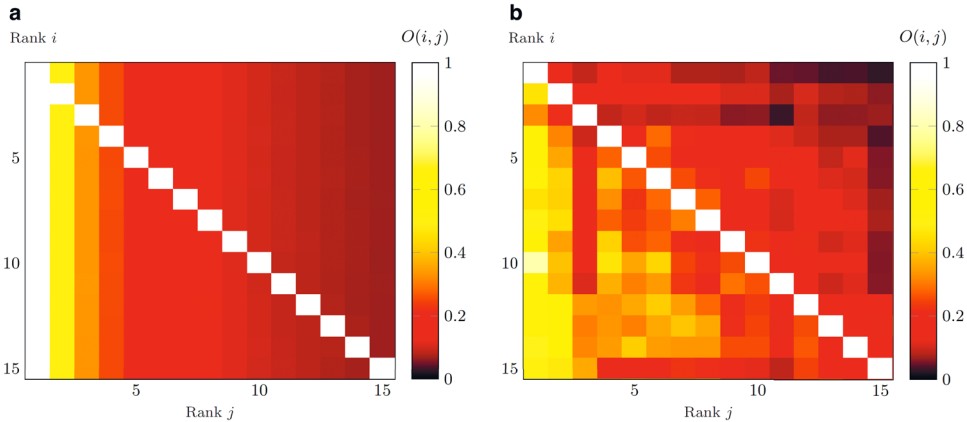

**Fig. 10 Followers' overlap analysis.** Followers' overlap results among the top 15 nodes. **a** The average numerical results were obtained from 10 simulations with $10^5$ agents, upon reaching equilibrium. **b** The results from the Twitch data set related to the chess category. An equivalent result for the poker data set is reported in Supplementary Fig. 14.

network size $N$, yet it remains above 10% on networks composed of $10^6$ nodes. Only a marginal increase is observed when introducing the preferential attachment on the meeting process (see the comparison in Supplementary Fig. 4).

**Audience overlap.** The previous analysis on the clustering coefficient studied the probability that two user's followees are friends with each other. Similarly, it is interesting to analyze the probability that two (highly followed) users are followed by the same third user. In other words, we aim at studying the similarity between the followers' sets of the different agents. The similarity between agents reveals the existence of common interest, and it can be used for link prediction[54] or to improve the recommendation systems[55]. Inspired by the Jaccard index introduced for species similarities[56], and already used to measure followers' overlap[57], we propose the following real-valued matrix $O$ to measure the overlap between the audiences of two agents:

$$O(i,j) := \frac{\left| \mathcal{F}_i^{\text{in}} \cap \mathcal{F}_j^{\text{in}} \right|}{\left| \mathcal{F}_i^{\text{in}} \right|} \in [0,1], \tag{10}$$

if $\left| \mathcal{F}_i^{\text{in}} \right| > 0$, and 0 otherwise, where $\mathcal{F}_i^{\text{in}}$ denotes the set of followers of agent $i$. In other words, this coefficient measures the number of common followers of $i$ and $j$, normalized by the number of followers of $i$. Note that, when agent $i$ is lower in the

ranking list with respect to agent $j$, i.e., $i > j$, we typically have $\left| \mathcal{F}_i^{\text{in}} \right| < \left| \mathcal{F}_j^{\text{in}} \right|$, and eq. (10) corresponds to the Szymkiewicz-Simpson coefficient, also known as the overlap coefficient[58], where the denominator is replaced by $min\{|\mathcal{F}_i^{\text{in}}|, |\mathcal{F}_j^{\text{in}}|\}$. Compared with it and to the Jaccard index[56], whose denominator is $|\mathcal{F}_i^{\text{in}} \cup \mathcal{F}_j^{\text{in}}|$, our measure leads to a non-symmetric matrix.

In Fig. 10a, we plot the numerical results on the overlap index for networks of $10^5$ nodes, upon reaching the equilibrium. The results are averaged over 10 simulations. According to the previous results, all the nodes should follow node 1 at equilibrium, thus any follower of a node $i$ should also be a follower of node 1 and the overlap in the first column is simply $(O(i,1) = 1)$. Moreover, consistent with our result in eq. (8), we observe the Zipf's sequence 1, 1/2, 1/3, …, in the first row $(O(1,j) = 1/j)$. Perhaps surprisingly, a similar pattern appears in all the rows, upon averaging on a sufficiently large number of simulations. We emphasize that the results are independent of the number of nodes $N$. An intuition for these observations is provided in Supplementary Fig. 5.

To further validate our theoretical model, we perform the same analysis on the overlap among the 15 most followed nodes of our Twitch data sets, reported in Fig. 10b (for the chess data set) and in Supplementary Fig. 14 (for the poker data set). Our numerical results are qualitatively well aligned with the real-world data with

respect to the horizontal decrease of the overlap index. Yet, the Twitch data sets show that the overlap index is not always uniform across the different rows. For instance, in the chess data set, low-ranking nodes exhibit a slightly higher overlap index with respect to the numerical results. Conversely, high-ranking nodes show the opposite behavior. This phenomenon might be related to the statistical dependency of individuals' followee sets on their out-degrees, as discussed in Supplementary Note 4.

## Discussion

Many of today's most popular online social networks are heavily based on UGC. Based on empirical evidence from longitudinal Twitter data, we proposed a meritocratic quality-based network formation model in which actors aim at optimizing the quality of the received content by strategically choosing their followees. We then analytically and numerically studied the properties of the resulting networks, in terms of in-degree and out-degree distributions, diameter, average clustering coefficient, and overlap among the followers' sets. In particular, we found that the meritocratic principle leads to Zipf's law of the expected in-degree as a function of the quality ranking. Remarkably, the result is robust against the effect of a preferential attachment-based meeting process (which mimics the recommendation systems). The theoretical predictions have been validated against empirical network data collected from Twitch.

Despite being of simple and parsimonious form, our model already captured several macroscopic features of today's UGC-based online social networks, e.g., scaling-free or small-world properties. Furthermore, thanks to its simplicity, the model can be extended in different directions, e.g., by considering different update rules or enriching it with well-known sociological incentives, e.g., network closure. Another possibility is to introduce multi-dimensional quality attributes, to cope with the possibility of multiple interests. This may lead to an interesting analysis of the competitive structural advantage of nodes with a diversified audience. The model can also be adapted to a growing network formation model, in which users join at different times, which will allow studying the rise (and eventually fall) of some network influencers. Another direction consists in the analysis of the network spreading characteristics, with particular emphasis on the influencers[59]. Ideally, this should be coupled with empirical analysis on different platforms, e.g., Instagram or Tik Tok, which are predominant among the new generations. In addition, longitudinal data could be used to make interesting predictions on the rise of new influencers. Last but not least, future work could extend our preliminary results on the role of the recommendation systems acting on the social media platforms as well as their effect on the users' behavior. The interplay between users' behavior and platforms mechanisms represents a widely unexplored research direction that may shed light on the effect of digitalization on our societies.

## Methods

**Theoretical analysis**. The goal of our theoretical analysis is to derive the macroscopic statistical properties of the network resulting from the quality-based network formation model. Our first result concerns the convergence to an equilibrium state, as in the following theorem.

**Theorem 4**. (Convergence) For any set of qualities $\{q_1, \ldots, q_N\}$, the network reaches an equilibrium almost surely. Moreover, the probability of reaching equilibrium within $t$ time-steps reads as follows:

$$\mathsf{P}[\text{An equilibrium is reached within } t \text{ time steps}]$$
$$= \mathsf{P}[a_{12}(t) = 1, a_{21}(t) = 1, \ldots, a_{N1}(t) = 1]$$
$$= \mathsf{P}[a_{12}(t) = 1] \times \mathsf{P}[a_{21}(t) = 1] \times \ldots \times \mathsf{P}[a_{N1}(t) = 1]$$
$$= \left(1 - \left(\frac{N-2}{N-1}\right)^t\right)^N.$$

**Proof**. Let $t > 0$ and $U(t)$ be a Bernoulli random variable such that $U(t) = 1$ if the network formation dynamics has reached equilibrium within $t$ time-steps, and 0 otherwise. As potential connections are uniformly randomly selected, the

probability that an agent $i \neq 1$ has not found agent 1 (or that agent 1 has not found agent 2) within $t$ time-steps is equal to $\left(\frac{N-2}{N-1}\right)^t$. Then, the probability that an agent has found her target within $t$ time-steps is complementary of the above, i.e., $1 - \left(\frac{N-2}{N-1}\right)^t$. Finally, since the dynamics of the individuals are independent of each other,

$$\mathsf{P}[U(t) = 1] = \left(1 - \left(\frac{N-2}{N-1}\right)^t\right)^N \to 1, \text{ as } t \to \infty.$$

The proofs of the results on the in-degree and out-degree probability distributions are discussed in Supplementary Note 1.

**Experimental setup**. In order to validate the statistical results of our quality-based model, we collected three data sets on Twitch, an online social media platform focusing on video streaming that recently became extremely popular among gamers. Twitch users can create their dedicated channels to stream their gameplay. Their UGC, in the form of live-streaming, can be browsed in appropriate categories corresponding to specific games. Thus, users can watch the streamed content of others, and eventually become followers.

Dealing with complex real-world networks poses several problems. In particular, systems are continuously changing not just in terms of network ties, but also with new nodes (users) joining and leaving the networks. To specifically validate our model results, we need (i) first to identify a suitable category of common interest, and (ii) second to reconstruct the social network among the users that show interest in this category. According to our modeling assumptions, the system is closed with respect to the set of users, and the network formation process is a consequence of users' interest in a specific topic. In the context of Twitch, this requires that the set of users interested in one (and only one) specific game, or topic, is fixed over time.

In order to minimize the chance of user's interest liability, we restricted our crawling setup to the users streaming and watching in one of the following three categories: chess, poker, and art (see Supplementary Fig. 8 for the analysis on the historical trend). Furthermore, we filtered our data by language, retaining only the English-speaking users that constitute the vast majority (see Supplementary Fig. 9). In this way, we avoid the possibility of multiple overlapping coherent sets (see Supplementary Note 3). Finally, we used an interest index to retain only those users that consistently stream in one of the chosen categories and to filter out those that may have accumulated audience because of streaming in other categories (see Supplementary Fig. 11). Based on the results of this criterion, we decided to exclude the data set related to the category art (see Supplementary Fig. 10 and related discussion in Supplementary Note 3).

We then set up the two Twitch data crawling on the categories chess and poker. On Twitch, not all the users provide their UGC, therefore nodes can be partitioned into two classes: broadcasters, i.e., users that provide UGC, and viewers. As the two partitions are heavily unbalanced, the network can approximately be considered as a quasi bipartite network, in which there are almost no ties among the viewers, very few ties (in absolute number) among the broadcasters, and most of the ties are directed from the viewers to the broadcasters. Note that this specific network structure, which we refer to as bipartite-like network, is compatible with our model predictions, see the analysis presented in Supplementary Note 1.

Due to the characteristics of the Twitch platform, the data crawling is necessarily limited to the users that are currently live-streaming. To overcome this limitation we repeated our crawling every hour for a period of 1 week (starting on 20 September 2020), until reaching a stable and consistent ranking in the list of the top 30 broadcasters. By doing so, though, our data may suffer from a sampling bias because we naturally tend to underestimate broadcasters that are not frequently active on the platform. Likely, these broadcasters are also not the most popular users. Fortunately, this sampling bias only affects the left tail of the in-degree distribution, but not the right tail in which we have the most influential nodes. We found 305 unique users streaming in the chess category, and 358 in the poker one. We then crawled the followers of these broadcasters obtaining a total of 690'917 (708'443) unique users and 1'450'403 (1'739'712) ties directed towards the broadcasters in the chess (poker) category. Finally, we reconstructed the bipartite-like networks of broadcasters and viewers as shown in Supplementary Fig. 7. Overall, the time-span of the network formation processes underlying the data sets is comprised between April 2010 and September 2020.

**Reporting summary**. Further information on research design is available in the Nature Research Reporting Summary linked to this article.

## Data availability

The Twitter and Twitch data sets collected in this study have been deposited at the ETH Research collection database. They are publicly available at the following repositories: https://doi.org/10.3929/ethz-b-000511049 (Twitter data set)[30] and https://doi.org/10.3929/ethz-b-000511065 (Twitch data sets)[36].

## Code availability

The code that performs the simulations and the analysis of the data sets is available at the following public repository https://doi.org/10.3929/ethz-b-000512497[60].

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

## Acknowledgements

The authors gratefully acknowledge financial support from ETH Zürich and National Natural Science Foundation of China under the grant number 72131001.

## Author contributions

N.P., W.M., and F.D. designed research; N.P. performed research; N.P. and C.L. collected data; N.P. and C.L. performed data analysis; N.P. wrote this manuscript; W.M and F.D. edited this manuscript.

## Competing interests

The authors declare no conflict of interest.

**Additional information**

