## [Peer Review File · Nature Communications]

REVIEWER COMMENTS

Reviewer #1 (Remarks to the Author):

* Most social network formation models are based on the setting where connections are formed based on popularity of nodes (preferential attachment model), an inherent fitness metric (Caldarelli, Capocci et al.) or the distance between nodes (Watts-Strogatz model). Varying from this paradigm, the authors propose a model where the connections are based on the quality of the content that is posted by the nodes; this is mostly the case in social networks such as Twitter and Instagram. Then, this proposed meritocracy based model is compared with empirical data and, some of its properties such as the forms of degree distributions are theoretically derived.

I think the key idea on which the proposed model is based is interesting and timely since most models that are widely used at the moment are not based on the quality of the user generated content (UGC). Also, the conclusions reached from its analysis are also interesting. However, I have several questions and comments regarding the results and the conclusions reached by the authors.

* According to authors, the quality of the UGC is the driving force of the link generation i.e. link forms as a consequence of the aim of the individuals to seek better content from the followees. But the popularity also affects the links (e.g. as a measure of the trustworthiness) Could this be changed to incorporate both the degree and UGC quality? For example, instead of uniformly picking a node at each time instant, this could reflect picking a node based on a probability distribution of both the degree and UGC quality or, there could be a biased coin flip at each time instant which indicate whether the nodes are chosen based on UGC quality (i.e., the index) or based on the degree. This would be a interpolation between regular preferential attachment and the UGC quality based method and would allow to explore it more carefully with respect to the bias of the coin flip. If this is a possibility, it would shed more light in to the novelty of the authors' idea.

* What happens if q_i is assigned from a non-uniform distribution? This seems more natural since quality of the content is not uniformly distributed in the real-world (more like an exponential distribution since the higher quality is very scarce in almost all cases).

* Related to the above, How is this approach different from the fitness model when the fitness parameter is used to reflect the quality of the UGC? More specifically, can this model be thought of as a special case (or vice versa) of:

Krioukov, D., Papadopoulos, F., Kitsak, M., Vahdat, A., & Boguná, M. (2010). Hyperbolic geometry of complex networks. *Physical Review E*, 82(3), 036106.

Caldarelli, Guido, et al. "Scale-free networks from varying vertex intrinsic fitness." *Physical review letters* 89.25 (2002): 258702.

* In the Twitch dataset, how do the authors identify the index (which correspond to the UGC quality)? (I tried to infer this from the paper and SI but failed to do figure it out). I think this should be stressed more as it is a key step in the empirical validation of the results.

* The authors mention that the in-degree distribution is not a power-law in both the smaller in-degree region as well as the tail of the distribution. While it makes intuitive sense why the smaller in-degree region is not a power-law, the fact that the tail portion is not a power-law is contradictory to the widely observed empirical evidence that the tail of (most) network degree distribution should follow a power-law. Could the authors clarify why this is reasonable?

* In addition to the above, the model proposed by the authors leads to an out-degree distribution which is not necessarily a power-law. However, it has been also observed that both the out-degree and in-degree distributions of directed networks are power-laws in many cases. Some of the early models (e.g.

Bollobás, B., Borgs, C., Chayes, J. T., & Riordan, O. (2003, January). Directed scale-free graphs. In SODA (Vol. 3, pp. 132-139.) are based on this fact.

The above is also evidence that preferential attachment is not necessarily limited to undirected graphs (slightly contradictory to the claim that "Unlike the preferential attachment model [36] which is based on undirected networks, our quality based model shows that scale-free in-degree distributions"). It would be a useful addition to the paper if the authors could clearly highlight why such a deviation might be useful and reasonable?

* Also, while the authors mention "...we showed that our theoretical out-degree distribution is similar to a log-normal distribution, with expectation equal to the harmonic number of the network size...", I could not find evidence to this claim in the paper. The non-monotonic nature in Figure 8 alone is not enough to claim that it is a log-normal distribution. The Clauset et al.'s approach maybe one way to see if log-normal is in fact a better fit (in a statistically significant sense) to the distribution and if so, what are it's parameters.

* Related to the above point, how well do the theoretically predicted out- and in- degree distributions of the proposed model match with those observed in the Twitch dataset that the authors use (and also, possibly a dataset based on the Twitter network that the authors frequently use as an example). I think this is an important point to address since only the in-degree vs. rank is illustrated empirically but, the degree distribution ultimately is the more practically relevant quantity.

* I am unable to understand the claim "Moreover, the result quantitatively matches the empirical observations on the small-world property of real-world networks, according to which the distance between two randomly chosen nodes grows proportionally to the logarithm of network size [59]." More specifically, how do the authors reach this conclusion via the degree distributions alone without the path length analysis (which is a crucial part of the small-world property). It would be great if the authors could clarify this claim a bit more and the underlying reasoning behind it.

Minor points/ corrections:

* I think "outdegree" should be "out-degree" throughout the paper. Similarly, the "indegree" should be "in-degree".

* "the highest quality node expects to have twice (three times) as many followers as the second (third) highest, and so on." - I think the term "respectively" should be used before the terms in brackets since authors use the brackets to clarify things further in all other instances except this one.

* In Fig. 1 caption, "Kolmogorov-Smirnof test" should be "Kolmogorov-Smirnov test".

* In Equation 7 of the corollary, the time variable t should not appear after taking its limit I think. Also, the indices "ij" have typos in that expression.

Reviewer #2 (Remarks to the Author):

The paper proposes an approach for network formation process.

The proposed approach is mainly based on the quality of user-generated content (UGC) associated with the nodes (users, agents), and which represents the likelihood the content will be liked by others. The proposed network formation process follows a stochastic process, where agents meet with uniform probability, and strategically create connections (directed ties) according to a utility maximization principle (based on the UGC quality, and rooted in an intuitive meritocratic principle.) When the process reaches an equilibrium, the resulting network, as the paper claims, exhibits:

- (1) an indegree distribution that follows a Zipf's law, and
- (2) an outdegree distribution that follows a log-normal distribution (with an expectation that equals the harmonic number of the network size).

Those results are validated by an empirical analysis of a real-world social network extracted from Twitch.

The proposed approach as well as the major findings are novel and interesting to others in the community and the wider field of social networks analysis. In particular, the introduction of the idea of associating a `quality` feature to agents into the network formation process.

While the overall work is pretty well done, several aspects should be considered:

1. While the proposed approach theoretically suits general (directed) networks, the empirical validation is conducted specifically on bipartite networks only (among two distinct sets of content providers and consumers).

This discrepancy raises a two-fold issue:

- Wouldn't the proposed approach be indeed more suitable for bipartite networks and hence shouldn't it be modeled as such?
- The empirical results might be mainly due to the fact that the (validation) networks are bipartite. In other words, being bipartite, the validation networks do not contain ties between the top influencers; if those absent ties existed, the results (i.e., degree-distributions) might not fit the theoretical model! Therefore, the empirical validation needs to be conducted using a general (not necessarily bipartite) networks, to prove the validity of the theoretical model.

2. In the proposed quality-based model, the average content quality of an agent is represented as a attribute that is drawn from the uniform probability distribution.

Why? I think a normal distribution of this attribute is more reasonable and realistic.

I think the paper needs a discussion of this choice, i.e., to compare the consequences of choosing a uniform- vs. a normal distribution on the theoretical and empirical findings.

3. Also in the proposed quality-based model, the authors assume that there exist no two agents with equal quality.

What happens when this assumption is not valid?

What are consequences of having agents with equal quality on the results?

4. Regarding Figure 4 (log-log plot) which shows empirical evidences of the Zipf's law, where it compares the actual data points (rank and indegree) and their linear regression fit. Although the slope (regression coefficient) and Pearson correlation are reported (both close to -1), the quality (accuracy) of the linear regression itself, in terms of R2 and/or RMSE, is missing!

This evaluation of "how accurate the linear regression" is essential to validate the claim that that figure shows empirical evidences of the Zipf's law.

Moreover, what happens if the plot is not log-log? what kind of fitting could be used instead? and what differences in the goodness-of-fit could be observed?

5. A further analysis, both theoretically and empirically, of the equilibrium state is needed. For instance, how many time steps are required to reach equilibrium?

Reviewer #3 (Remarks to the Author):

I studied the manuscript "A meritocratic network formation model for the rise of social media influencers" by Pagan et al.

The paper scope seems to be very narrow to me - the authors consider

1) an outdated model, that is the one of Ref.(24) cited in the bibliography, with a minimal change on the choice of the agent fitness (or "quality rank" in their jargon) represented by Eq.(2) in page 3 of the paper, that amounts to a trivial change of the function MAX() into the function AVERAGE() as compared to the existing ten-years-old model. The subsequent calculations seems to me a standard application. The contribution to the field, to the best of my ability to judge, is marginal --The authors explicitly state in their paper, that: "Both definitions, though, share the same meritocratic principle"--

2) They also check numerically that their calculation is valid in a network of $O(10^3)$ nodes, which is nowhere near to the size of real world social networks where "influencers" rise and operate. Their benchmark system is roughly 6-order-of-magnitude smaller than typical social media platform (e.g. Instagram).

3) The sociological context provided is very weak in my opinion, and thus i remain unconvinced even about the importance of the problem they tackled, let alone the solution they eventually provide.

4) Can Instagram or Tik Tok be described using this method?

5) Does the "meritocratic principle" really help explaining the quality of UGC? How about the role of profiling and artificial custom audience in the decision making process of human users?

5) Alternatively, does your approach solve an open problem in the field of machine learned models of human behavior?

I urge the authors to see my criticism in a positive manner i.e. to add to the introduction more motivation and to add at the end a discussion of their results e.g. by comparing them to real data (and not a cherry-picked network formed by a vanishing subset of users of Twitter or Twitch), rather than simulations or alternatively to explain the relevance of their model and corrections to an actual experiment i.e. to make a falsifiable prediction about a yet unobserved social network (for example about the rise of a yet undiscovered influencer and check if she becomes one and if so when).

Statement of Revision

We greatly thank the editorial board for giving us the opportunity to improve our manuscript and submit a revised version. We also would like to thank all the reviewers for their constructive and insightful comments, which we have carefully addressed in the revised manuscripts. Our major changes can be summarized as follows.

- (i) We entirely revise our introduction by highlighting the motivation and significance of our work, as well as adding some extra references suggested by the reviewers;
- (ii) We extend our theoretical analysis to explain the formation of a specific type of bipartite-like networks, which resemble the underlying structure of some most popular online platforms nowadays, e.g., YouTube and Twitch;
- (iii) We extend our numerical analysis to the study of important network properties such as the diameter and the clustering coefficient;
- (iv) We also study the effect of the recommendation systems, which have an increasingly relevant role in today's platforms, by incorporating the well-established preferential-attachment mechanism in our network formation process. Our analysis shows that the meritocratic principle (based on the quality) dominates the effect of the recommendation systems (which only modifies the "opportunity" of being found, but not the "choice" of following).

In the revised manuscript, all the revisions are marked blue. We greatly appreciated the quality of the reviews, which pointed us to many interesting extensions. In the following, we provide a detailed answer for each of the points raised by the reviewers.

Comments by Reviewer # 1

[R1: 1] " MOST SOCIAL NETWORK FORMATION MODELS ARE BASED ON THE SETTING WHERE CONNECTIONS ARE FORMED BASED ON POPULARITY OF NODES (PREFERENTIAL ATTACHMENT MODEL), AN INHERENT FITNESS METRIC (CALDARELLI, CAPOCCI ET AL.) OR THE DISTANCE BETWEEN NODES (WATTS-STROGATZ MODEL). VARYING FROM THIS PARADIGM, THE AUTHORS PROPOSE A MODEL WHERE THE CONNECTIONS ARE BASED ON THE QUALITY OF THE CONTENT THAT IS POSTED BY THE NODES; THIS IS MOSTLY THE CASE IN SOCIAL NETWORKS SUCH AS TWITTER AND INSTAGRAM. THEN, THIS PROPOSED MERITOCRACY BASED MODEL IS COMPARED WITH EMPIRICAL DATA AND, SOME OF ITS PROPERTIES SUCH AS THE FORMS OF DEGREE DISTRIBUTIONS ARE THEORETICALLY DERIVED. I THINK THE KEY IDEA ON WHICH THE PROPOSED MODEL IS BASED IS INTERESTING AND TIMELY SINCE MOST MODELS THAT ARE WIDELY USED AT THE MOMENT ARE NOT BASED ON THE QUALITY OF THE USER GENERATED CONTENT (UGC). ALSO, THE CONCLUSIONS REACHED FROM ITS ANALYSIS ARE ALSO INTERESTING. HOWEVER, I HAVE SEVERAL QUESTIONS AND COMMENTS REGARDING THE RESULTS AND THE CONCLUSIONS REACHED BY THE AUTHORS. "

We greatly thank Reviewer #1 for the careful reading and positive evaluation of our work. Thanks to her/his constructive comments, the revised version includes several improvements.

[R1:2] “ ACCORDING TO AUTHORS, THE QUALITY OF THE UGC IS THE DRIVING FORCE OF THE LINK GENERATION I.E. LINK FORMS AS A CONSEQUENCE OF THE AIM OF THE INDIVIDUALS TO SEEK BETTER CONTENT FROM THE FOLLOWEES. BUT THE POPULARITY ALSO AFFECTS THE LINKS (E.G. AS A MEASURE OF THE TRUSTWORTHINESS) COULD THIS BE CHANGED TO INCORPORATE BOTH THE DEGREE AND UGC QUALITY? FOR EXAMPLE, INSTEAD OF UNIFORMLY PICKING A NODE AT EACH TIME INSTANT, THIS COULD REFLECT PICKING A NODE BASED ON A PROBABILITY DISTRIBUTION OF BOTH THE DEGREE AND UGC QUALITY OR, THERE COULD BE A BIASED COIN FLIP AT EACH TIME INSTANT WHICH INDICATE WHETHER THE NODES ARE CHOSEN BASED ON UGC QUALITY (I.E., THE INDEX) OR BASED ON THE DEGREE. THIS WOULD BE A INTERPOLATION BETWEEN REGULAR PREFERENTIAL ATTACHMENT AND THE UGC QUALITY BASED METHOD AND WOULD ALLOW TO EXPLORE IT MORE CAREFULLY WITH RESPECT TO THE BIAS OF THE COIN FLIP. IF THIS IS A POSSIBILITY, IT WOULD SHED MORE LIGHT IN TO THE NOVELTY OF THE AUTHORS’ IDEA. ”

Thank you for your comment and suggestion. Following up on your idea, we performed several numerical experiments. Concretely, though, rather than using a biased coin to regulate the link decision, we considered integrating a preferential attachment mechanism in the *meeting process*. In the previous version of our model, the meeting process was based on a uniform probability density function such that every user had the same probability of finding any other user. In the revised version, we studied a meeting process in which users’ probability to be met is proportional to their current in-degree, as in the preferential attachment model [1]. By doing so, we try to mimic the effect of the recommendation systems which may increase the probability of being exposed to the content of high in-degree nodes. On the other hand, though, we leave the original individual linking decision-making untouched, i.e., agent i follows a new agent j only if the quality of j increases the utility function of agent i (which measures the maximum quality of the content i is exposed to through his/her followees). Note that, by doing so, we also make sure that the dynamics converge (almost surely) to an equilibrium state, whose properties can be analyzed and compared. On the contrary, with the method suggested by the reviewer, we would have to artificially stop the dynamics after a given number of time-steps (to be arbitrarily defined), because the preferential attachment process does not terminate. We also believe that our modification better represents the real world.

To quantify the effect of this preferential attachment based meeting process, we compared three different scenarios. In the first one, we used a uniform probability distribution, in the second one, we equally mixed uniform probability distribution and preferential attachment, and in the third one, we only used preferential attachment (based on the in-degree of the nodes). Since a theoretical analysis of the second and third scenarios is increasingly complex, we perform a Monte Carlo analysis and run 1000 simulations (per scenario) and generate the corresponding equilibrium networks.

On the left column of SR fig. 1, we present the results on the rank vs in-degree plot. Noticeably, introducing the (full or mixed) preferential attachment process increases the variance in the in-degree probability distribution of each agent. For instance, especially in the third scenario, it is not unlikely that some of the high-quality nodes (e.g., in the top 50) receive less than 10 followers (which, instead, has almost zero probability in the uniform distribution scenario). In other words, the preferential attachment may penalize some agents, which receive fewer followers than what they would expect with the uniform distribution process (whose theoretical average corresponds to the Zipf’s law). On the other hand, it is also possible that some low-quality agents get an initial (i.e., in the early stages of our network formation process) advantage (purely by chance) which gets reinforced by the preferential attachment mechanism. Yet, this effect quickly fades away, because potential followers still undergo the quality threshold rule. Interestingly, though, the average number of followers as a function of the ranking is close to the Zipf’s law, even in the case of full or mixed preferential attachment. The coefficient of the fitted line goes from -1 in the uniform distribution case, to -0.94 in the pure preferential attachment. Moreover, it is remarkable that, on average, the meritocratic principle remains satisfied even after introducing the preferential attachment process. In fact, while the increasing variance of the probability distribution may more frequently result in

SR Figure 1. Numerical results of 1000 simulations with 1000 nodes for three different meeting process scenarios (top: uniform distribution, bottom: pure preferential attachment, middle: mixed). On the left, the color maps show the in-degree probability density function, as a function of the quality rank. In orange, the average in-degree. In blue, the Zipf's law which equals the expected in-degree in the uniform distribution scenario. In purple, the linear fit (in log-log) of the average values. On the right, the numerical probability density functions (orange) and their power-laws (blue) and log-normal (purple) fits. In black, the theoretical results for the scenario with uniform probability distribution is plot as reference.

lower quality nodes having higher in-degree than higher quality ones, the correlation between quality and followers persists: the higher the quality, the higher the average number of followers.

On the right column of SR fig. 1, we also plot the average in-degree probability distribution functions for the three scenarios. Together with them, we plot (in black) the theoretical results obtained for the first scenario (the uniform distribution), and the best power-law and log-normal fits. The major difference between the scenarios is on the right heavy tail. While the uniform distribution meeting process predicts (with lower variance) the in-degree of top quality nodes, i.e., the peaks located at $d = N/2, N/3, \dots$, in the pure preferential attachment process there is no Zipf’s law. Yet, the log-normal and power-law fits of the three scenarios are very similar in terms of their parameters. Interestingly, the mixed scenario is more similar to the uniform distribution rather than to the pure preferential attachment, even though the probability of choosing a potential followee from the uniform distribution or from the preferential attachment process is equally split.

To complement our analysis on the effect of the preferential attachment meeting process, in SR fig. 2 we study the numerical out-degree distributions for the three scenarios considering 1000 network realizations of 1000 nodes. From the comparison, we evince that the preferential attachment slightly shifts the probability density function to the left, but the shape of the distribution remains unvaried.

SR Figure 2. Numerical results of the empirical out-degree distribution of 1000 simulations of equilibrium networks of 1000 nodes, for the three different scenarios: uniform distribution, pure preferential attachment process, and mixed approach.

Finally, we also studied the impact of the preferential attachment based meeting process on the number of time-steps required to achieve equilibrium as well as on the average clustering coefficient (see [R2:6]). In this regard, the introduction of the preferential attachment process significantly speeds up the convergence to the equilibrium state, and it slightly increases the average clustering coefficient.

In summary, we found that the effect of changing the meeting process to moderate preferential attachment (i.e., with 50% probability from uniform distribution and with 50% probability from preferential attachment) is minor with respect to the (individual and average) in-degree distributions, as well as to the out-degree distribution, and the average clustering coefficient. The impact is more noticeable (especially in the individual and average in-degree distributions) only when using a pure preferential attachment.

Based on our responses above, in the revised manuscript, we mention the possibility of having multiple scenarios, and we add a numerical analysis of the number of time-steps needed to reach an equilibrium state, for the three scenarios we mentioned above. Furthermore, we introduce a section named “Preferential

attachment meeting process” in which we discuss some of the differences between the scenario with the uniform distribution and the one with the mixed preferential attachment in-degree based. Finally, we report the detailed analysis related to the in-degree and out-degree distribution in Supplementary Note 2.

[R1:3] “ WHAT HAPPENS IF q_i IS ASSIGNED FROM A NON-UNIFORM DISTRIBUTION? THIS SEEMS MORE NATURAL SINCE QUALITY OF THE CONTENT IS NOT UNIFORMLY DISTRIBUTED IN THE REAL-WORLD (MORE LIKE AN EXPONENTIAL DISTRIBUTION SINCE THE HIGHER QUALITY IS VERY SCARCE IN ALMOST ALL CASES). ”

Thank you for this comment, which allows us to clarify an important aspect. The quality parameters q_i can be drawn from any distribution, without this affecting the results. The reason behind this lies in the fact that only the ordering of the quality attributes, but not their magnitude, is driving the dynamics. In fact, agent i 's individual decision of following a new agent j only depends on the comparison between the quality of j , q_j , and the quality of i 's followees, as prescribed in the equations (2) and (3). Since we order the agents according to their quality attribute, it is sufficient that j is smaller than k , for all k in i 's followees' set. Therefore, our model is independent of any numerical representation of “quality”. This is indeed a desired feature since the quality of a content usually does not have a numerical nature and its quantification could be arbitrary.

Nevertheless, we agree with the reviewer that a uniform distribution for the quality attribute is potentially unrealistic, and other distributions, e.g., an exponential distribution (or a normal distribution, as suggested by Reviewer #2 in [R:2.3]), might be more appropriate. Thus, we decided to revise the manuscript including the following comment:

Similarly to the approach in the fitness model [2], we endow each actor i with an attribute q_i , drawn from a probability distribution, e.g., uniform, normal, exponential distribution, that describes the average quality of i 's content [3], e.g., a picture taken at that traveling destination. As will be manifested later, our model predictions are independent of the numerical representation of these qualities, which could be somehow subjective and arbitrary. Instead, in our model, only the ordering of the individual qualities matters. Therefore, the choice of the underlying probability distribution does not affect any of the following results, contrary to the fitness model [2].

[R1:4] “ RELATED TO THE ABOVE, HOW IS THIS APPROACH DIFFERENT FROM THE FITNESS MODEL WHEN THE FITNESS PARAMETER IS USED TO REFLECT THE QUALITY OF THE UGC? MORE SPECIFICALLY, CAN THIS MODEL BE THOUGHT OF AS A SPECIAL CASE (OR VICE VERSA) OF: KRIOUKOV, D., PAPADOPOULOS, F., KITSAK, M., VAHDAT, A., & BOGUNÁ, M. (2010). HYPERBOLIC GEOMETRY OF COMPLEX NETWORKS. PHYSICAL REVIEW E, 82(3), 036106. CALDARELLI, GUIDO, ET AL. "SCALE-FREE NETWORKS FROM VARYING VERTEX INTRINSIC FITNESS." PHYSICAL REVIEW LETTERS 89.25 (2002): 258702. ”

We thank the reviewer for pointing us to this relevant reference [2]. The fitness parameter in the fitness model plays a similar role when compared to our quality attribute. However, we would like to emphasize some of the differences between the fitness model and our quality-based model.

- In the fitness model, the fitness parameters are derived from a predefined probability distribution (similarly to our quality attribute). Yet, the scale-free property emerging in the fitness model is related to particular types of distributions of the fitness parameter, e.g., another scale-free distribution (even though the authors show that this is not a unique choice [2, 4]). On the other hand, our results are independent of the probability distribution from which qualities are drawn (see our response to [R1:3]). We highlighted this difference including the following comment in the revised manuscript:

As will be manifested later, our model predictions are independent of the numerical representation of these qualities, which could be somehow subjective and arbitrary. Instead, in our model, only the ordering of the individual qualities matters. Therefore, the choice of the underlying probability distribution does not affect any of the following results, contrary to the fitness model [2].

- The fitness model is proposed for undirected networks, thus the probability of a link between i and j is proportional to both the fitness of i and j . In our quality-based model, we focus on directed networks in which agents are interested to follow good-quality nodes, thus only the quality of the potential new followee (and the quality of the current followees) is relevant. This poses more attention to the individual decision-making process.
- In the fitness model, there is no utilitarian principle which drives the network formation. In our model, we assume users aim at maximizing the received content quality, thus they continuously look for increasing-quality agents. The same principle incorporates the individuals' willingness to best using their limited time available on social network platforms by making non-profitable connections infinitely costly. On the other hand, the increasing-quality property of the connection sequences is not present in the fitness model (as well as in the preferential attachment model, as emphasized in the manuscript). As a matter of fact, with the fitness model, it is more likely that a node connects to other nodes with decreasing fitness over time. This is contradictory to our analysis of the Twitter data-set which shows that users look for increasing quality connections, as well as to the need for high-quality content proposed by content aggregators and other "repost" accounts. We highlighted this difference including the following comment in the revised manuscript: *Moreover, even in the fitness model [2] where the quality is considered, users tend to connect to high quality nodes first, rather than later.* Furthermore, without such a utilitarian principle, the fitness model (similarly to the preferential attachment model), does not have a sociologically principled stopping criterion: the number of connections each node creates must be predefined.
- Our results, not only are compatible with the widely observed scale-free property, but they are consistent with a Zipf's law distribution of the in-degree with respect to the quality ranking. To the best of our knowledge, the fitness model does not lead to similar conclusions.
- Finally, our model exhibits other network properties, e.g., out-degree distribution, overlap coefficient, clustering coefficient, network diameter, which are compatible with real-world networks.

However, we agree that our quality-based model shares the same principle that "the good-get-richer", as mentioned in [5], therefore we decided to revise the manuscript including a short review in the introduction

However, except for the fitness model [2] in which users are connected with probability proportional to the individuals' fitness attributes, the large multidisciplinary interest in the study of network formation has so far privileged topological and socio-economic aspects observed in offline social networks (or in online social networks which mimic them, e.g., Facebook) and neglected the effect of the UGC.

[R1:5] " IN THE TWITCH DATA-SET, HOW DO THE AUTHORS IDENTIFY THE INDEX (WHICH CORRESPOND TO THE UGC QUALITY)? (I TRIED TO INFER THIS FROM THE PAPER AND SI BUT FAILED TO DO FIGURE IT OUT). I THINK THIS SHOULD BE STRESSED MORE AS IT IS A KEY STEP IN THE EMPIRICAL VALIDATION OF THE RESULTS. "

Thank you for this comment. The UGC qualities are key underlying parameters in our proposed model. It is the UGC qualities that determine whether a user decides to follow another user. In general, it is very challenging to propose a widely accepted measure of UGC qualities since they are not numerical by nature. Instead of first quantifying all the users' UGC qualities and then directly comparing the users' UGC qualities with their numbers of followers, in this manuscript, we validate our model in an indirect way. That is, we validate our model's macroscopic and statistical predictions by empirical data. For example, our proposed model predicts that the resulting network's in-degree distribution follows the Zipf's law, and we indeed observe the presence of Zipf's in the Twitch data set crawled from the Internet, see fig. 4 in the main text. Such consistency suggests that our model could be a reasonable explanation of the UGC-based network formation process.

SR Figure 3. The scatter plot between the proposed measure of quality, i.e., the average views per hour for each user’s broadcasts, and the in-degrees, i.e., the numbers of followers, for the top 50 influencers on Twitch on the topic “chess”. The blue straight line is the corresponding linear regression.

In addition to the above results presented in our manuscript, we also tried proposing a measure of the UGC qualities that is accessible from the aforementioned Twitch data set. We use the average number of views per hour to characterize the qualities of the video broadcasts on Twitch. Then, for each Twitch user (broadcaster), we compute the average quality of all the broadcasts they contribute, and we plot the scatter plot between such “user qualities” and their number of followers, see SR fig. 3. These results indicate that users’ qualities (in the above sense) and their number of followers are indeed highly correlated. However, since such measurement of qualities is not beyond any reasonable doubt, we do not include these results in our manuscript.

[R1:6] “ THE AUTHORS MENTION THAT THE IN-DEGREE DISTRIBUTION IS NOT A POWER-LAW IN BOTH THE SMALLER IN-DEGREE REGION AS WELL AS THE TAIL OF THE DISTRIBUTION. WHILE IT MAKES INTUITIVE SENSE WHY THE SMALLER IN-DEGREE REGION IS NOT A POWER-LAW, THE FACT THAT THE TAIL PORTION IS NOT A POWER-LAW IS CONTRADICTIONARY TO THE WIDELY OBSERVED EMPIRICAL EVIDENCE THAT THE TAIL OF (MOST) NETWORK DEGREE DISTRIBUTION SHOULD FOLLOW A POWER-LAW. COULD THE AUTHORS CLARIFY WHY THIS IS REASONABLE? ”

Thank you for your comment. First of all, we would like to point out that we do not claim that our distribution does not have power-law features. As a matter of fact, fig. 5 (in the manuscript) shows that the power-law is a good fit, especially in the central part of the plot. On the other hand, certain oscillations around the power-law baseline can be observed in the right tail of the distribution. These are the result of a more accurate description of the in-degree of the top-ranking nodes: according to our theoretical results, the expected in-degree of the nodes follows a Zipf’s law, with relatively low variance (see fig. 3 in the manuscript). The effect of this high precision (low variance) in the probability distribution functions of the top in-degree nodes determines the peaks observable in fig. 5 in the manuscript, as well as in SR fig. 4. On the other hand, if we use the canonical (yet for our scenario arguably too coarse) logarithmic binning of the data, we obtain the gray data points in SR fig. 4, in which oscillations become smooth to the point that they follow very closely the power-law fit (in blue). Note that the process of logarithmic binning the data is usually applied because, in the right tail, data are so scattered that need to be binned together in order to show the desired power-law behavior. In our paper, instead, we purposefully highlight the differences,

because our model not only captures the overall power-law trend but also predicts (with high precision), the in-degree of top-ranking nodes, which follows (in expectation) a Zipf’s law.

SR Figure 4. Probability density function of the nodes’ in-degree in a network of $N = 1000$ agents. In orange, we plot the numerical distribution resulting from 1000 simulations upon reaching equilibrium. In gray, we plot the numerical distribution upon using logarithmic binning, as common practice when showing power-law features. In blue, we fit the numerical data with a power-law of coefficient $\hat{\alpha} = -2.055 \pm 0.0028$ using the algorithm in Clauset et al. [6].

Based on our response, in fig. 5 in the revised manuscript we also show the data after using the classical logarithmic binning, and we added the following comment:

In synthesis, our probability distribution differs from Pareto and log-normal distributions when focusing on the network influencers. In this range, i.e., in the extreme right tail of the distribution, our theoretical distribution is similar to its power-law fit, only after logarithmically (and thus coarsely) binning the data (as common practice when data are scattered and the distributions are continuous).

[R1: 7] “ IN ADDITION TO THE ABOVE, THE MODEL PROPOSED BY THE AUTHORS LEADS TO AN OUT-DEGREE DISTRIBUTION WHICH IS NOT NECESSARILY A POWER-LAW. HOWEVER, IT HAS BEEN ALSO OBSERVED THAT BOTH THE OUT-DEGREE AND IN-DEGREE DISTRIBUTIONS OF DIRECTED NETWORKS ARE POWER-LAWS IN MANY CASES. SOME OF THE EARLY MODELS (E.G. BOLLOBÁS, B., BORGS, C., CHAYES, J. T., & RIORDAN, O. (2003, JANUARY). DIRECTED SCALE-FREE GRAPHS. IN SODA (VOL. 3, PP. 132-139).) ARE BASED ON THIS FACT. THE ABOVE IS ALSO EVIDENCE THAT PREFERENTIAL ATTACHMENT IS NOT NECESSARILY LIMITED TO UNDIRECTED GRAPHS (SLIGHTLY CONTRADICTORY TO THE CLAIM THAT “UNLIKE THE PREFERENTIAL ATTACHMENT MODEL [36] WHICH IS BASED ON UNDIRECTED NETWORKS, OUR QUALITY BASED MODEL SHOWS THAT SCALE-FREE IN-DEGREE DISTRIBUTIONS”). IT WOULD BE A USEFUL ADDITION TO THE PAPER IF THE AUTHORS COULD CLEARLY HIGHLIGHT WHY SUCH A DEVIATION MIGHT BE USEFUL AND REASONABLE? ”

Thank you for this comment. We do agree that the out-degree distributions in many real-world networks obey power-laws, e.g., the “internet graphs” as mentioned in the paper by Bollobás et al. [7]. We also agree that the preferential attachment model is not limited to undirected graphs and many extensions are available. However, respectfully, we believe that, for a large class of social networks, the out-degree distributions do not necessarily obey the power-law. Evidence for this is found on, e.g., YouTube [8] or Instagram [9, 10]. Rather, several studies [11, 12, 13] have witnessed the presence of (at least) two different

scaling regimes which are observed in the out-degree distribution of many online social networks. In particular, they highlight a much faster decrease (quantifiable in terms of the higher magnitude of the scaling coefficient) in the probability of seeing very high out-degree nodes, thus in the right heavy tail of the distribution. This result is not surprising: take YouTube, Instagram, or Twitch as examples. Some influencers might have one million followers since it costs “nothing” to be followed by someone. On the contrary, it is hard to imagine that some YouTube, Instagram, or Twitch user follows one million other users (even ten thousand is not a realistic number), since the action of following someone involves a decision-making process and at least a click on the “Follow” button, which takes some time and thus makes it implausible to follow a huge amount of other users. The only exception to this argument is constituted by potentially fake-users accounts, which can artificially create a very large number of out-going connections. To prevent the spread of such accounts, many online platforms adopted the solution to limit the maximum number of out-connections. As a result, the out-degree distribution of these platforms features a clear cut-off on the order of roughly thousand of connections. Therefore, for such online social networks, one should not necessarily expect the out-degree distributions to follow power-laws. Rather, the out-degree distribution might exhibit a faster decrease (compared to a power-law) in its heavy tail. Such behavior is potentially compatible with our out-degree distribution, which is plotted (in log-log scale) in SR fig. 5. More detailed analysis of our distribution will be discussed in the next comment [R:1.8].

SR Figure 5. Out-degree probability density function (left) and complementary cumulative distribution (right) for different size N of the network.

Based on our responses above, in the revised manuscript, we clarify the applicability of the preferential attachment model to both undirected and directed graphs. We also justify that not all social networks should obey the power-laws in terms of their out-degree distributions. We refer to the “Out-degree distribution” section in the revised manuscript for the details.

[R1:8] “ ALSO, WHILE THE AUTHORS MENTION “... WE SHOWED THAT OUR THEORETICAL OUT-DEGREE DISTRIBUTION IS SIMILAR TO A LOG-NORMAL DISTRIBUTION, WITH EXPECTATION EQUAL TO THE HARMONIC NUMBER OF THE NETWORK SIZE...”, I COULD NOT FIND EVIDENCE TO THIS CLAIM IN THE PAPER. THE NON-MONOTONIC NATURE IN FIGURE 8 ALONE IS NOT ENOUGH TO CLAIM THAT IT IS A LOG-NORMAL DISTRIBUTION. THE CLAUSET ET AL.’S APPROACH MAYBE ONE WAY TO SEE IF LOG-NORMAL IS IN FACT A BETTER FIT (IN A STATISTICALLY SIGNIFICANT SENSE) TO THE DISTRIBUTION AND IF SO, WHAT ARE IT’S PARAMETERS. ”

We thank the reviewer for their comment. As a matter of fact, our out-degree distribution is statistically different from both a power-law and a log-normal distribution. Yet, it has some similarities with the log-normal distribution with regards to its non-monotonic nature and its very fast decrease in the right tail. However, this decrease, better shown in log-log scale in SR fig. 5, is more compatible with that of a gamma distribution (or a Poisson distribution) rather than that of a log-normal, as shown in SR fig. 6. According to the results of the Kolmogorov-Smirnov test, the distance between the empirical distribution and the gamma, Poisson, log-normal, and power-law distributions are, respectively, 0.095, 0.17, 0.49, and 0.97.

SR Figure 6. Results of 1000 simulations of a network of 1000 nodes: on the left, the out-degree probability density function, on the right the complementary cumulative distribution function. The Gamma distribution is best fitted with parameters: $\alpha = 28.66$, $loc = -5.46$, $\beta = 0.45$. The Poisson distribution is fitted with parameter λ equal to the Harmonic number of $N - 1$, the power-law with parameters $\alpha = -12.17 \pm 0.068$, $d_{min} = 13$, and the log-normal with parameters $\mu = 1.96$, $\sigma = 0.35$.

The behavior of a gamma (or a Poisson) distribution is compatible with the cut-off (artificial or not) which is present in empirical out-degree distribution, e.g., on Instagram [9, 10] or YouTube [8]. Moreover, the non-monotonic feature is also compatible with the fact that the majority of (active) users on these social networks have at least (and at most) $10^1 - 10^2$ out-connections, as shown in the empirical Instagram data-sets of [9, 10]. In other words, the out-degree Complementary Cumulative Distribution function is substantially flat until reaching $d = 10^1 - 10^2$, as shown in SR fig. 5. We also emphasize that our network formation model is focused on the ties that are created due to a certain interest. Clearly, users have multiple interests, thus their actual out-degree observed in the social networks (in the order of approximately 10^2 connections) is the result of the superposition of different “common-interest” subnetworks of the kind that we are modeling.

Based on our responses above, in the revised manuscript, we clarify the similarities with the log-normal distribution and we add the discussion on the gamma and Poisson distributions.

Even though the theoretical out-degree distribution cannot be directly associated to any known distribution, it is very close to a gamma distribution (or to a Poisson distribution), as shown in Supplementary fig. 2 (see also the discussion in Supplementary Note 2). Thanks to its non-monotonic nature and its fast decay, there are also some similarities with the log-normal distributions (whose decrease is more than linear in the log-log plot).

Further details are also available in the revised version of our Supplementary Note 2, in the section named “Out-degree distribution”.

[R1:9] “ RELATED TO THE ABOVE POINT, HOW WELL DO THE THEORETICALLY PREDICTED OUT- AND IN- DEGREE DISTRIBUTIONS OF THE PROPOSED MODEL MATCH WITH THOSE OBSERVED IN THE TWITCH DATA-SET THAT THE AUTHORS USE (AND ALSO, POSSIBLY A DATA-SET BASED ON THE TWITTER NETWORK THAT THE AUTHORS FREQUENTLY USE AS AN EXAMPLE). I THINK THIS IS AN IMPORTANT POINT TO ADDRESS SINCE ONLY THE IN-DEGREE VS. RANK IS ILLUSTRATED EMPIRICALLY BUT, THE DEGREE DISTRIBUTION ULTIMATELY IS THE MORE PRACTICALLY RELEVANT QUANTITY. ”

Thank you for your comment. First, we would like to emphasize that the Zipf’s law shown on the in-degree vs. rank plot is more suitable for the analysis on the scaling of the high in-degree nodes. In fact, while a consistent body of literature has focused on the properties of the in-degree distribution as a whole, little is known on the scaling relation between the most influential nodes of the network. In this region, power-laws and log-normal provide a continuous approximation of the data by (logarithmically; see our response to your earlier comment [R1:6]) binning the nodes together, yet it is unclear whether they exhibit Zipf’s (discrete) behavior. In our work, we focus our attention on the emergence of scaling between these nodes. Thus we rather focus on the in-degree vs ranking relation. However, we agree with the reviewer on the importance of evaluating and comparing theoretical and empirical results on the in-degree distribution.

In SR fig. 7 (now also reported in Supplementary fig. 12) we complete our analysis by showing the histograms of the Complementary Cumulative (top) and probability density (bottom) empirical distributions for the chess data-set (similar results are obtained for the poker data-set). We would like to emphasize that the empirical in-degree distribution is likely to be affected by a sampling bias due to the fact that we can only fetch the broadcasters that were actively broadcasting during our data collection period. While we used a sufficiently large time-window which guaranteed us that the list of the 30 most followed broadcasters remained invariant for a certain number of fetches, we might still have missed a portion of low-to-medium-popular broadcasters. We are certainly aware of this limitation, yet we did not consider it to be a major issue, because such a sampling bias is not affecting the most popular nodes.

The log-normal fit (in orange) is a good approximation when focusing on the low in-degree nodes region. On the other hand, the log-normal distribution deviates substantially from the empirical data in the high in-degree nodes region. We also used two different power-laws to fit the empirical data. In purple, a power-law of parameter $\alpha = 1.72$ (and $d_{min} = 1425$) is the best fit (in terms of log-likelihood). While it provides a good description of the middle-ranking nodes, its slope is not a good fit for the high in-degree nodes. A slightly better power-law fit can be obtained by forcing $d_{min} = 18000$ (blue line). In this case, the coefficient is very close to $\alpha = 2$, and the fit captures the scaling of the high in-degree nodes. Finally, in black we plot a pure Zipf’s law, in which each node has in-degree proportional to the inverse of its ranking. In this case, the Zipf’s law correctly captures the trend of the high in-degree nodes, which tend to follow the sequence $N, N/2, N/3, \dots$. Note that the pure Zipf’s law used is a computationally tractable approximation of our theoretical results: in our analysis, each node follows the Zipf’s law *only* in expectation. Yet, such a distribution is a good approximation of our theoretical probability distribution function in the region of high in-degree nodes, as shown in SR fig. 8, while it deviates marginally in the region of low in-degree nodes.

With respect to the out-degree distributions, the empirical results of the two Twitch data-sets are reported in SR fig. 9, and in the manuscript in fig. 8. Both empirical distributions exhibit fat tails, yet, compared to the tails of the in-degree distribution (shown in SR fig. 7), they have a much faster decay. As a result, the maximum out-degree is more than three orders of magnitude smaller than the maximum in-degree. Both empirical distributions, though, also show some discrepancies with respect to the theoretical distribution:

- the left tail of the empirical distribution is heavier than the theoretical one. In other words, in the empirical distribution the share of nodes with small out-degree, e.g., $d^{out} = 1, 2, \dots$ is significantly larger than the predicted one. The reasons for this could be either the recommendation systems (as

SR Figure 7. Comparison between empirical distribution of the in-degree probability distribution function of the chess data-set and log-normal, power-law, and Zipf’s law fit. On top, the Complementary Cumulative distribution function, while on bottom the probability density function are displayed. The log-normal fit has parameters $\mu = 5.57, \sigma = 2.40$. The power-law fit (in purple) has parameters $\alpha = 1.72, \sigma = 0.08, d_{\min} = 1425$. The power-law fit with $d_{\min} = 18000$ (in blue) has parameters $\alpha = 1.95, \sigma = 0.25$. Finally, the Zipf’s law distribution is a computationally tractable approximation of our theoretical distribution, as shown in SR fig. 8.

SR Figure 8. Comparison of the Complementary Cumulative distribution functions of our theoretical distribution (with uniform probability in the meeting process) and a pure Zipf’s law, for a network with $N = 1000$ nodes.

explained in the manuscript), but it could also depend on the sampling bias explained previously. In our data-sets, we only fetched the broadcasters that were active during our data collection, thus we only have a lower bound of the out-degree of the followers (by collecting more broadcasters, we could have found that the same follower was following more broadcasters). Preliminary numerical results show that this feature can be achieved by means of a growing random network version (in which nodes are added sequentially) of our model.

- the right tail is not decreasing as fast as the predicted one (on the other hand, is decreasing much faster than a typical power-law distribution, as shown in the SR fig. 9. One possible explanation for this may be found in the different linking behavior of a minority of users. In fact, some users may have less restrictive policies in accepting new followees. Preliminary simulation results show that relaxing the dynamics by using the average received quality instead of the maximum quality as a threshold for a new followee naturally shifts the out-degree distribution towards higher frequencies of high out-degree nodes. Furthermore, some users might strategically use their “Follow” action in the hope of being reciprocated. Taking into account a larger spectrum of linking behavior (possibly also strategic) may increase the goodness-of-fit of the empirical distributions.

For the reasons exposed so far, we believe that further investigation is needed on the out-degree distribution. While our work is currently limited by the sampling bias, we also believe that the model can be enriched to capture, e.g., heavier tails compared to those predicted by our simil-exponential distribution.

Based on our responses above, in the revised Supplementary Information, we added the analysis and comparison of the empirical in-degree distribution, and we refer to it from the manuscript:

We finally refer to Supplementary Note 4 (see in particular Supplementary figs. 12, 13) for a detailed analysis of the empirical in-degree distribution.

SR Figure 9. Empirical distributions of the out-degree probability density function of the two Twitch data-sets.

Furthermore, we extensively revised our analysis on the empirical out-degree distribution, and we added a comparison with our theoretical results, as well as with a power-law distribution.

[R1: 10] “ I AM UNABLE TO UNDERSTAND THE CLAIM “MOREOVER, THE RESULT QUANTITATIVELY MATCHES THE EMPIRICAL OBSERVATIONS ON THE SMALL-WORLD PROPERTY OF REAL-WORLD NETWORKS, ACCORDING TO WHICH THE DISTANCE BETWEEN TWO RANDOMLY CHOSEN NODES GROWS PROPORTIONALLY TO THE LOGARITHM OF NETWORK SIZE [59].” MORE SPECIFICALLY, HOW DO THE AUTHORS REACH THIS CONCLUSION VIA THE DEGREE DISTRIBUTIONS ALONE WITHOUT THE PATH LENGTH ANALYSIS (WHICH IS A CRUCIAL PART OF THE SMALL-WORLD PROPERTY). IT WOULD BE GREAT IF THE AUTHORS COULD CLARIFY THIS CLAIM A BIT MORE AND THE UNDERLYING REASONING BEHIND IT. ”

Thank you for this comment. To clarify our statement, in SR fig. 10 (now also reported in fig. 9 in the manuscript) we provide a numerical analysis of the average network diameter and average nodes’ distance for different network sizes.

According to the numerical results, the network diameter has a growth rate similar to the base-2 logarithm of the network size. This indicates that the resulting network features the small-world network property. Moreover, the average node’s distance has even slower growth, and it results in the order of 3 for a network of 2000 nodes. Unfortunately, we reach computational complexity thresholds when computing the diameter of larger networks. Nonetheless, we believe that larger networks will feature the same small-world property. Yet, further investigation would be useful. Finally, we would like to emphasize that the results are also quantitatively similar to previous findings on an Instagram network sample [14] (of size equal to 44 thousand nodes) where the diameter is found to be 11, and the average distance 3.16.

Based on our responses above, in the revised manuscript, we provided an extra section named “Diameter and clustering” in which we clarify our statement and we added the numerical analysis just presented.

[R1: 11] “MINOR COMMENTS: ”

- “ I THINK “OUTDEGREE” SHOULD BE “OUT-DEGREE” THROUGHOUT THE PAPER. SIMILARLY, THE “INDEGREE” SHOULD BE “IN-DEGREE”. ”

Thank you for the suggestion. We revised the manuscript accordingly.

SR Figure 10. Numerical analysis of the average diameter and average nodes’ distance for different network sizes N . For each value of N , we run 100 simulations. The shaded area indicates one standard deviation from the average value. For comparison, we plot in gray the base 2 logarithm of the network size.

- “ “THE HIGHEST QUALITY NODE EXPECTS TO HAVE TWICE (THREE TIMES) AS MANY FOLLOWERS AS THE SECOND (THIRD) HIGHEST, AND SO ON.” - I THINK THE TERM “RESPECTIVELY” SHOULD BE USED BEFORE THE TERMS IN BRACKETS SINCE AUTHORS USE THE BRACKETS TO CLARIFY THINGS FURTHER IN ALL OTHER INSTANCES EXCEPT THIS ONE. ”

Thank you for the suggestion. We revised the manuscript accordingly.

- “ IN FIG. 1 CAPTION, “KOLMOGOROV-SMIRNOF TEST” SHOULD BE “KOLMOGOROV-SMIRNOV TEST”. ”

Thank you for the correction. We revised the manuscript accordingly.

- “ IN EQUATION 7 OF THE COROLLARY, THE TIME VARIABLE T SHOULD NOT APPEAR AFTER TAKING ITS LIMIT I THINK. ALSO, THE INDICES “IJ” HAVE TYPOS IN THAT EXPRESSION. ”

Thank you for the correction. We revised the manuscript accordingly.

We finally thank the reviewer again for all their constructive comments which prompted a lot of changes in the paper and led to an improved manuscript.

Comments by Reviewer # 2

[R2: 1] “ THE PAPER PROPOSES AN APPROACH FOR NETWORK FORMATION PROCESS. THE PROPOSED APPROACH IS MAINLY BASED ON THE QUALITY OF USER-GENERATED CONTENT (UGC) ASSOCIATED WITH THE NODES (USERS, AGENTS), AND WHICH REPRESENTS THE LIKELIHOOD THE CONTENT WILL BE LIKED BY OTHERS. THE PROPOSED NETWORK FORMATION PROCESS FOLLOWS A STOCHASTIC PROCESS, WHERE AGENTS MEET WITH UNIFORM PROBABILITY, AND STRATEGICALLY CREATE CONNECTIONS (DIRECTED TIES) ACCORDING TO A UTILITY MAXIMIZATION PRINCIPLE (BASED ON THE UGC QUALITY, AND ROOTED IN AN INTUITIVE MERITOCRATIC PRINCIPLE.) WHEN THE PROCESS REACHES AN EQUILIBRIUM, THE RESULTING NETWORK, AS THE PAPER CLAIMS, EXHIBITS: (1) AN INDEGREE DISTRIBUTION THAT FOLLOWS A ZIPP’S LAW, AND (2) AN OUTDEGREE DISTRIBUTION THAT FOLLOWS A LOG-NORMAL DISTRIBUTION (WITH AN EXPECTATION THAT EQUALS THE HARMONIC NUMBER OF THE NETWORK SIZE). THOSE RESULTS ARE VALIDATED BY AN EMPIRICAL ANALYSIS OF A REAL-WORLD SOCIAL NETWORK EXTRACTED FROM TWITCH. THE PROPOSED APPROACH AS WELL AS THE MAJOR FINDINGS ARE NOVEL AND INTERESTING TO OTHERS IN THE COMMUNITY AND THE WIDER FIELD OF SOCIAL NETWORKS ANALYSIS. IN PARTICULAR, THE INTRODUCTION OF THE IDEA OF ASSOCIATING A ‘QUALITY’ FEATURE TO AGENTS INTO THE NETWORK FORMATION PROCESS. WHILE THE OVERALL WORK IS PRETTY WELL DONE, SEVERAL ASPECTS SHOULD BE CONSIDERED: ”

We greatly thank Reviewer #2 for their comments, careful reading, and positive evaluation of our work. Their suggestions have prompted a number of interesting investigations which have been integrated in the manuscript and in the Supplementary Information.

[R2: 2] “ WHILE THE PROPOSED APPROACH THEORETICALLY SUITS GENERAL (DIRECTED) NETWORKS, THE EMPIRICAL VALIDATION IS CONDUCTED SPECIFICALLY ON BIPARTITE NETWORKS ONLY (AMONG TWO DISTINCT SETS OF CONTENT PROVIDERS AND CONSUMERS). THIS DISCREPANCY RAISES A TWO-FOLD ISSUE: - WOULDNT’ THE PROPOSED APPROACH BE INDEED MORE SUITABLE FOR BIPARTITE NETWORKS AND HENCE SHOULDN’T IT BE MODELED AS SUCH? - THE EMPIRICAL RESULTS MIGHT BE MAINLY DUE TO THE FACT THAT THE (VALIDATION) NETWORKS ARE BIPARTITE. IN OTHER WORDS, BEING BIPARTITE, THE VALIDATION NETWORKS DO NOT CONTAIN TIES BETWEEN THE TOP INFLUENCERS; IF THOSE ABSENT TIES EXISTED, THE RESULTS (I.E., DEGREE-DISTRIBUTIONS) MIGHT NOT FIT THE THEORETICAL MODEL! THEREFORE, THE EMPIRICAL VALIDATION NEEDS TO BE CONDUCTED USING A GENERAL (NOT NECESSARILY BIPARTITE) NETWORKS, TO PROVE THE VALIDITY OF THE THEORETICAL MODEL. ”

Thank you for raising these questions and our responses are summarized as follows and elaborated afterwards.

- (a) Empirical evidence suggests that the underlying network among Twitch users is not perfectly bipartite, since there exist some links among the influencers;
- (b) However, the Twitch network does exhibit some bipartite-like features;
- (c) The bipartite-like structure of the Twitch network is compatible with the predictions by our quality-based model;
- (d) In fact, if needed, our quality-based model can be slightly adapted to generate a perfectly bipartite network, which still preserves the in-degree and out-degree distributions predicted by the original model.

Firstly, we would like to point out that, rigorously speaking, the underlying network among Twitch users is not a bipartite graph such that all the links are from followers to influencers. Empirical evidence shows that there exist a non-negligible number of interconnections among the top influencers. We analyze the data set for all the active broadcasters (492 in total), i.e., the influencers, on the topic “chess”. These influencers have 13’203 out-links linked to other influencers. Although maybe not all the links among the influencers are built based on the meritocratic principle, the inter-influencer links cannot be simply neglected. In fact,

the interconnections among the influencers are even denser than the link density among all the Twitch users interested in chess. The link density, i.e.,

$$\frac{\text{number of directed links}}{\text{maximal number of possible directed links}}$$

for the subgraph induced by the influencers is around 5%, while the link density for the entire network including both the influencers and the followers is as low as $1.73 * 10^{-6}$. The above empirical evidence clearly indicates that the Twitch network is effectively not a bipartite graph where there are only links from followers to influencers.

That being said, the links between the influencers constitute less than 1% of the ties that are directed to them (13'203, as just reported, out of 1'450'403, as reported in the manuscript). This is due to the fact that the partitions between influencers and followers are heavily unbalanced. Thus, even though the Twitch network is not exactly a bipartite network, it still exhibits some bipartite-like structure. Namely, there are links among the influencers, but most of the links, instead, are from the followers to the influencers.

We would like to clarify that, although our quality-based model applies to general directed graphs, by assigning zero quality to the followers and positive qualities to the influencers, our model leads to predictions compatible with the aforementioned special bipartite-like structure of the Twitch network. To theoretically establish this point, we include a new section named "Special case: Bipartite-like networks" in Supplementary Note 1. Furthermore, in the manuscript, we added the following comment in the Methods section:

We then set up the two Twitch data crawling on the categories chess and poker. On Twitch, not all the users provide their UGC, therefore nodes can be partitioned into two classes: broadcasters, i.e., users that provide UGC, and viewers. Since the two partitions are heavily unbalanced, the network can approximately be considered as a quasi bipartite network, in which there are almost no ties among the viewers, very few ties (in absolute number) among the broadcasters, and most of the ties are directed from the viewers to the broadcasters. Note that this specific network structure, which we refer to as "bipartite-like" network, is compatible with our model predictions, see the analysis presented in Supplementary Note 1.

On the other hand, our model can be slightly adapted so that it generates perfectly bipartite networks, in which all the links are from followers to influencers. From the analysis in the section "Special case: Bipartite-like networks" of Supplementary Note 1, one could easily infer that, if the followers are assigned zero quality and the influencers are not allowed to follow other influencers, then such a slight modification will lead to a perfectly bipartite network with only links from followers to influencers. Suppose there are m influencers and n followers. Then, in such a perfectly bipartite network, the in-degree distribution for the influencers still satisfies equations (5)-(8) in the main text, with $N = n$, while the out-degree distribution for the followers satisfies equation (9) in the main text, with $N = m + 1$. The only minor exception is that the most popular influencer has the in-degree equal to n instead of $n - 1$.

[R2: 3] "IN THE PROPOSED QUALITY-BASED MODEL, THE AVERAGE CONTENT QUALITY OF AN AGENT IS REPRESENTED AS A ATTRIBUTE THAT IS DRAWN FROM THE UNIFORM PROBABILITY DISTRIBUTION. WHY? I THINK A NORMAL DISTRIBUTION OF THIS ATTRIBUTE IS MORE REASONABLE AND REALISTIC. I THINK THE PAPER NEEDS A DISCUSSION OF THIS CHOICE, I.E., TO COMPARE THE CONSEQUENCES OF CHOOSING A UNIFORM- VS. A NORMAL DISTRIBUTION ON THE THEORETICAL AND EMPIRICAL FINDINGS. "

Thank you for this comment, which allows us to clarify an important aspect. The quality parameters q_i can be drawn from any distribution, without this affecting the results. The reason behind this lies in the fact that only the ordering of the quality attributes, but not their magnitude, is driving the dynamics. In fact, agent i 's individual decision of following a new agent j only depends on the comparison between the

quality of j , q_j , and the quality of i 's followees, as prescribed in the equations (2) and (3). Since we order the agents according to their quality attribute, it is sufficient that j is smaller than k , for all k in i 's followees' set. Therefore, the predictions of our model are independent of any numerical representation of "quality". Such independence is a desirable feature since the quality of content usually does not have a numerical nature and its quantification could be arbitrary.

Nevertheless, we agree with the reviewer that a uniform distribution for the quality attribute is potentially unrealistic, and other distributions, e.g., a normal distribution (or an exponential distribution, as suggested by Reviewer #1 in [R1:3]), might be more appropriate. Thus, we decided to revise the manuscript including the following comment:

Similarly to the approach in the fitness model [2], we endow each actor i with an attribute q_i , drawn from a probability distribution, e.g., uniform, normal, exponential distribution, that describes the average quality of i 's content [3], e.g., a picture taken at that traveling destination. As will be manifested later, our model predictions are independent of the numerical representation of these qualities, which could be somehow subjective and arbitrary. Instead, in our model, only the ordering of the individual qualities matters. Therefore, the choice of the underlying probability distribution does not affect any of the following results, contrary to the fitness model [2].

[R2: 4] " ALSO IN THE PROPOSED QUALITY-BASED MODEL, THE AUTHORS ASSUME THAT THERE EXIST NO TWO AGENTS WITH EQUAL QUALITY. WHAT HAPPENS WHEN THIS ASSUMPTION IS NOT VALID? WHAT ARE CONSEQUENCES OF HAVING AGENTS WITH EQUAL QUALITY ON THE RESULTS? "

Thank you for your comment. If two agents have the same quality the difference in the results is marginal, and it only requires a bit more care (essentially an "if-then-else" case in the proof) in the theoretical analysis. However, in what follows we provide a brief numerical analysis, which should shed some light on the issue. To account for equal-quality users, we consider (and compare) two scenarios: in the first scenario (i), users follow the same dynamics as the one expressed in equation (3) in the manuscript. In this case, if there are 2 or more users with the same quality, following one of them prevents from following the other(s). In the second scenario (ii), we consider a modified dynamics, i.e.,

$$a_{ij}(t + 1) = \begin{cases} 1, & \text{if } q_j \geq V_i(t), \\ a_{ij}(t), & \text{otherwise,} \end{cases}$$

in which we introduce a non-strict inequality condition. Such a modification does not yield any change when the users have all different qualities. However, now, if there are 2 or more users with the same quality, following one of them does not prevent following the other(s). We tested the two scenarios on 10'000 simulations of a network of $N = 100$ nodes in which the quality of the nodes 3 and 4 is forced to be the same. Similarly, the quality of nodes 6, 7, and 8 is the same.

The results of the two scenarios are presented in SR fig. 11. First of all, we notice that the other nodes are not affected, and their in-degree probability density functions (as well as their average in-degree) is the same as in the situation without equal-quality nodes, i.e., correspond to the Zipf's law. On the other hand, the in-degree probability density functions are similar across agents with the same quality. The same applies to their average in-degree, thus the meritocratic principle is preserved. The only difference between the two scenarios is as follows: in (i), the average in-degree equals the Zipf's value computed at the lowest ranking node, i.e., $N/4$ for nodes 3 and 4, and $N/8$ for nodes 6, 7, and 8. Conversely, in (ii), the average in-degree is $N/3$ for nodes 3 and 4, and $N/6$ for nodes 6, 7, and 8. In summary, the nodes of equal quality remain on the left (respectively, right) of the Zipf's law in the first (respectively, second) scenario, while all other nodes satisfy the Zipf's law.

To conclude, we believe that it can be considered unlikely that two users have exactly the same quality.

SR Figure 11. Results for the equal quality test. On the left, we tested the (i) scenario, on the right the (ii) scenario, as described in [R2:4]. For both scenarios, we plot the empirical in-degree probability density functions (as a function of the node rank) resulting from 10'000 simulations (upon reaching convergence) for networks of $N = 100$ nodes. For this test, $q_3 = q_4$ and $q_6 = q_7 = q_8$. In blue, we plot the Zipf's law N/i , which corresponds to the expected in-degree when no equality nodes are present.

Nonetheless, with the above example we have illustrated that the difference is marginally small and limited to the prediction of those users.

Based on our responses above, in the revised manuscript, we added the following comment:

Yet, we emphasize that the model and our analysis can be generalized to the case where two or more agents have the same quality (see the example in Supplementary Note 2).

In the Supplementary Information, we then provide the example just mentioned.

[R2:5] “ REGARDING FIGURE 4 (LOG-LOG PLOT) WHICH SHOWS EMPIRICAL EVIDENCES OF THE ZIPF’S LAW, WHERE IT COMPARES THE ACTUAL DATA POINTS (RANK AND INDEGREE) AND THEIR LINEAR REGRESSION FIT. ALTHOUGH THE SLOPE (REGRESSION COEFFICIENT) AND PEARSON CORRELATION ARE REPORTED (BOTH CLOSE TO -1), THE QUALITY (ACCURACY) OF THE LINEAR REGRESSION ITSELF, IN TERMS OF R2 AND/OR RMSE, IS MISSING! THIS EVALUATION OF "HOW ACCURATE THE LINEAR REGRESSION" IS ESSENTIAL TO VALIDATE THE CLAIM THAT THAT FIGURE SHOWS EMPIRICAL EVIDENCES OF THE ZIPF’S LAW. MOREOVER, WHAT HAPPENS IF THE PLOT IS NOT LOG-LOG? WHAT KIND OF FITTING COULD BE USED INSTEAD? AND WHAT DIFFERENCES IN THE GOODNESS-OF-FIT COULD BE OBSERVED? ”

We thank the reviewer for their comment and we agree that the accuracy of the linear regression needs to be reported. Computing them for both data-sets, we obtained a value of $R^2 = 0.96$, $RMSE = 0.16$ (chess) and $R^2 = 0.95$, $RMSE = 0.17$ (poker). The R^2 and $RMSE$ measures, now reported in the caption of the fig. 4 of the revised manuscript, show that the Zipf’s law is a good fit for both data-sets. The same holds also when we plot the data in non log-log plot (left) or lin-log plot (right), as shown in SR fig. 12. For comparison, we also consider an exponential fit for both data-sets, which has worse residual performances compared to the linear log-log fit. In fact, in these cases, we obtain a value of $R^2 = 0.93$, $RMSE = 0.21$ (chess) and $R^2 = 0.91$, $RMSE = 0.22$ (poker).

SR Figure 12. Comparison of different fits for the rank vs in-degree data of the chess and poker data-set. On the left, the plot is shown with linear axis, on the right, with log axis in the y direction.

Finally, we believe that visualizing the data in log-log plot allows for an intuitive interpretation of the scaling law underlying the data, thus we did not change them in the manuscript.

[R2: 6] “ A FURTHER ANALYSIS, BOTH THEORETICALLY AND EMPIRICALLY, OF THE EQUILIBRIUM STATE IS NEEDED. FOR INSTANCE, HOW MANY TIME STEPS ARE REQUIRED TO REACH EQUILIBRIUM? ”

Thank you for this comment. Following the reviewer’s request, we investigated the number of time-steps needed to reach the equilibrium state. Furthermore, we decided to complement our analysis with the study of other network properties such as the network diameter and the clustering coefficient. In what follows, we present a summary of our results, which have also been integrated in the revised versions of the manuscript and the Supplementary Information.

First, we obtained the following theoretical results: the probability of reaching an equilibrium within t time-steps is as follows:

$$\begin{aligned} P[a_{12}(t) = 1, a_{21}(t) = 1, \dots, a_{N1}(t) = 1] &= P[a_{12}(t) = 1] \times P[a_{21}(t) = 1] \times \dots \times P[a_{N1}(t) = 1] = \\ &= \left(1 - \left(\frac{N-2}{N-1} \right)^t \right)^N, \end{aligned}$$

where we used the fact the dynamic of each agent is independent from the others, and that the probability that each agent finds its target within t time-steps is equal to $1 - ((N-2)/(N-1))^t$. From this cumulative distribution function, we can compute the expected number of time-steps to reach equilibrium:

$$\mathbb{E} = \sum_{t=1}^{\infty} t \left(P[a_{12}(t) = 1, a_{21}(t) = 1, \dots, a_{N1}(t) = 1] - P[a_{12}(t-1) = 1, a_{21}(t-1) = 1, \dots, a_{N1}(t-1) = 1] \right).$$

In SR fig. 13 we report (in blue) the above theoretical result, as well as the numerical results from 1000 simulations, as a function of the network size.

SR Figure 13. Numerical analysis of the average time-step for different network sizes N . For each value of N , we run 1000 simulations, in three different settings for the meeting process probability distribution: in blue, we use a uniform distribution, in orange, we use a preferential attachment mechanism, and in purple we use a mixed distribution (50% chance of uniform distribution and 50% change of Preferential attachment, for each meeting). The data-points indicate the median value. The shaded area indicates the first and third quantile.

For comparison, we have also considered other two scenarios (discussed in our response to [R1:2] by the first reviewer) that affect the choice of the meeting partner. In the standard scenario (blue), a uniform distribution process is used to propose new potential followees. In orange, instead, we use a preferential attachment approach for the meeting process. With the term “preferential attachment” it is indicated a specific procedure in which a vertex receives more edges according to its degree. In practice, at each time-step, each agent i meets a distinct agent j according to a probability which is proportional to the current in-degree of the agent j . By doing so, we combine the quality-based dynamics (in the “follow /not follow” decision) with a meeting process which resembles the effect of the recommendation systems. Finally, in purple, the meeting agent is chosen with 50% probability, from the uniform distribution, and with the remaining 50% probability, according to the preferential attachment mechanism. From the analysis of the results we conclude that the number of time-steps grows almost exponentially in the network size, when considering a uniform distribution process. Conversely, the pure preferential attachment based meeting process significantly reduces the number of required time-steps, even when it is mixed (at 50% probability) with a uniform distribution. Intuitively, the reason behind this lies in the fact that, with a uniform distribution process, it takes (on average) $N/2$ time-steps for an agent to find the best-quality nodes, even after having found the second best quality node. On the other hand, using a (full or mixed) preferential attachment meeting process, significantly speeds up the last part of the process, because high-ranking nodes (which ultimately are expecting to receive a high number of followers), are met with high probability.

Furthermore, we extended the analysis of the network properties at equilibrium in two different directions. In particular, first, we studied the diameter and average distance of the nodes as a function of the network size. The results, reported in our response to comment [R1:10] by the first reviewer, show that the resulting

networks exhibit small-world feature which are compatible with previous analysis on, e.g., Instagram [14]. Second, we studied the average clustering coefficient. Since we are dealing with directed networks, there are eight different configurations between each set of three connected nodes, which can be classified according to the direction of the ties as in [15]. According to their taxonomy, we opted for considering the class “out”, in which a triad around node i is considered closed when i follows two distinct agents j and k , and there exists a tie from j to k or from k to j . In other words, the clustering coefficient of node i reads as follows:

$$c_i = \frac{\sum_{j \neq i} \sum_{k \neq i, j} a_{ij} a_{ik} a_{jk}}{\sum_{j \neq i} \sum_{k \neq i, j} a_{ij} a_{ik}}$$

where a_{ij} denotes the tie from i to j . The results of the average clustering coefficient are shown in SR fig. 14, for the same three different scenarios of the meeting process. For each network size, the average is taken across 1000 simulations. While a monotonic decrease of the average clustering coefficient can be noticed for all three scenarios, only minor differences are noticeable across the scenarios. Ultimately, in the presence of (full or partial) preferential attachment, slightly larger values of average clustering coefficients are observed.

Average clustering

SR Figure 14. Numerical analysis of the average clustering coefficient for different network sizes $N = 10, 20, 30, 50, 70, 100, 150, 200, 500, 1000$. For each value of N , we run 1000 simulations, in three different settings for the meeting process probability distribution: in blue, we use a uniform distribution, in orange, we use a preferential attachment mechanism, and in purple we use a mixed distribution (50% chance of uniform distribution and 50% change of Preferential attachment, for each meeting). The data-points indicate the average value. The shaded area indicates one standard deviation.

Results for larger network sizes are reported only for the uniform distribution meeting process in SR fig. 15 (now also reported in fig. 9 in the manuscript), up to 20 thousand nodes. While the average clustering coefficient is monotonically decreasing, it still remains above 10% on networks composed of 10^6 nodes, which is consistent with empirical observations of Twitter or Instagram, as reported in the manuscript.

Based on our responses above, in the revised manuscript, we added the theoretical and numerical analysis of the number of time-steps by modifying the statement of our Convergence theorem and by providing the plot shown in SR fig. 13. Furthermore, in the newly added section “Diameter and clustering”, we

SR Figure 15. Numerical analysis of the average clustering coefficient for networks of different sizes. The data-points indicate the average value. The shaded area indicates one standard deviation.

present our analysis on the diameter and clustering coefficient. Part of this discussion is also addressed in the revised version of the Supplementary Information (see the “Clustering coefficient” section in Supplementary Note 2).

We finally thank the reviewer again for all their constructive comments which prompted a lot of changes in the paper and led to an improved manuscript.

Comments by Reviewer # 3

[R3: 1] “ I STUDIED THE MANUSCRIPT "A MERITOCRATIC NETWORK FORMATION MODEL FOR THE RISE OF SOCIAL MEDIA INFLUENCERS" BY PAGAN ET AL. ”

We greatly thank Reviewer #3 for taking the time for evaluating our work. Their suggestions have in particular led to a thorough revision of our introduction. Furthermore, our numerical analysis has now been extended to networks of size up to 10^7 nodes. More detail comments will follow.

[R3: 2] “ THE PAPER SCOPE SEEMS TO BE VERY NARROW TO ME - THE AUTHORS CONSIDER 1) AN OUTDATED MODEL, THAT IS THE ONE OF REF.(24) CITED IN THE BIBLIOGRAPHY, WITH A MINIMAL CHANGE ON THE CHOICE OF THE AGENT FITNESS (OR "QUALITY RANK" IN THEIR JARGON) REPRESENTED BY EQ.(2) IN PAGE 3 OF THE PAPER, THAT AMOUNTS TO A TRIVIAL CHANGE OF THE FUNCTION MAX() INTO THE FUNCTION AVERAGE() AS COMPARED TO THE EXISTING TEN-YEARS-OLD MODEL. THE SUBSEQUENT CALCULATIONS SEEMS TO ME A STANDARD APPLICATION. THE CONTRIBUTION TO THE FIELD, TO THE BEST OF MY ABILITY TO JUDGE, IS MARGINAL –THE AUTHORS EXPLICITLY STATE IN THEIR PAPER, THAT: "BOTH DEFINITIONS, THOUGH, SHARE THE SAME MERITOCRATIC PRINCIPLE"– ”

Thank you for your comment. Respectfully, we would like to explain why we do not think that the contribution of our paper amounts to a minimal change with respect to [3] (which corresponds to Ref.(24) in the reviewer’s comment). In fact, the motivation behind the two papers is profoundly different. We aim at introducing the idea of User-Generated Content (and of the quality associated to it) in the field which studies the process of social network formation. Most of the network formation literature has considered “rich-get-richer” effects, or sociological incentives, e.g., reciprocity or network closure, but has so far neglected the User-Generated Content, which on the other hand has become the life-blood of many of the new-generation online platforms. Furthermore, we are interested in studying the properties of the resulting network, and in verifying them against real-world social networks.

On the contrary, the authors of [3] are interested in game-theoretical mechanism aiming at incentivizing User-Generated Content contributions of higher quality in platforms such as Wikipedia. In order to approach this problem, they also introduced the concept of “quality” $q_j \in [0, 1]$ associated to the User-Generated Content of j , and assumed that viewers (followers) benefit from the quality of the content they receive in the form of an utility function proportional to

$$U_i = K^{-\alpha} \sum_{j=1}^K q_j,$$

for some $\alpha \in]0, 1[$. When $\alpha \rightarrow 1$, the utility function of the agent i is then proportional to the average quality, thus users aim at maximizing the average quality received. Yet, besides using a similar utility function for the viewers (or followers, in our case), they do not provide any network formation model, nor are interested in the properties of the resulting network. Ultimately, the scope of their paper is to provide a “game-theoretical model within which the design and performance of mechanisms for incentivizing high-quality User-Generated Content can be analyzed” [3]. Therefore, while our paper certainly shares some modeling assumptions with the framework proposed in [3], it significantly differs from it in terms of motivation, application, and results.

[R3: 3] “ THEY ALSO CHECK NUMERICALLY THAT THEIR CALCULATION IS VALID IN A NETWORK OF $O(10^3)$ NODES, WHICH IS NOWHERE NEAR TO THE SIZE OF REAL WORLD SOCIAL NETWORKS WHERE "INFLUENCERS" RISE AND OPERATE. THEIR BENCHMARK SYSTEM IS ROUGHLY 6-ORDER-OF-MAGNITUDE SMALLER THAN TYPICAL SOCIAL MEDIA PLATFORM (E.G. INSTAGRAM). ”

Thank you for your comment and we agree that the numerical analysis shown in the previous version

of the manuscript referred to networks which are considerably smaller than a typical real-world social network. However, we would like to emphasize that most of our theoretical results, e.g., the analysis of the in-degree and out-degree distributions, are valid for any network size, thus not necessarily restricted to networks of $O(10^3)$ nodes. Yet, for computational or illustration purposes, the plots were limited to moderately small network sizes.

During our revision, we managed to optimize the numerical computation and obtain simulation results for networks of size as large as 10^6 - 10^7 nodes (see the analysis on the out-degree probability density function in our response to comment [R1:8] or on the clustering coefficient in [R2:6]) which show the applicability of our model beyond the previous numerical limitations. The analysis has now been integrated in the revised versions of the manuscript (see in particular the sections on the out-degree distribution, and the new one on the diameter and clustering) as well as of the Supplementary Information. Interestingly, many of the network characteristics which have been analyzed, e.g., Zipf's law of the expected in-degree, out-degree probability distribution, overlap coefficient, network diameter, average node distance, and average clustering coefficient, are relatively robust with respect to the network size. In particular, the resulting networks always exhibit the most common features of real world social networks, i.e., a scaling law, the small-world effect (small diameter), and non-negligible clustering coefficient.

[R3: 4] “ 3) THE SOCIOLOGICAL CONTEXT PROVIDED IS VERY WEAK IN MY OPINION, AND THUS I REMAIN UNCONVINCED EVEN ABOUT THE IMPORTANCE OF THE PROBLEM THEY TACKLED, LET ALONE THE SOLUTION THEY EVENTUALLY PROVIDE. ”

Thank you for your comment. We conduct a major revision of the introduction to better highlight the motivation and significance of our work. In brief, we make the following points in the revised introduction:

- (a) In the past few years, online social networks, such as Instagram and Tiktok, are evolving into new forms and exhibit noticeably distinguishing features compared with some earlier online platforms, e.g., Facebook and LinkedIn. Two key distinguishing features are the large proportion of unilateral links and the prevalence of user generated content (UGC) [16, 9]. These features can have profound impacts on the structure of nowadays online social networks and many dynamic processes occurring on them [17, 18], as well as online business [19], which are largely based on the rise of new “internet influencers” [20, 21];
- (b) Based on the above observations, it is important to understand how the UGC relates with the emergence of online influencers and the resulting online social network structure;
- (c) Despite the UGC being a fundamental ingredient of nowadays social media platforms, its key feature, i.e., the quality of UGC, has not been incorporated in most previous network formation dynamics models [1, 22, 23, 24, 25, 26]. As a result, some delicate new features, e.g., some fine structure, of today's new online social platforms remain to be unveiled;
- (d) The importance of UGC and the lack of extensive studies on this topic inspire us to propose a novel network formation mechanism that incorporates both the utility maximization principle and the UGC qualities. This new model is simple in form yet leads to non-trivial quantitative predictions that are supported by empirical data.

Based on the arguments above, respectfully, we believe that our model has strong sociological context and is well-motivated. The motivation and significance of our work are now better highlighted in the revised introduction.

[R3: 5] “ CAN INSTAGRAM OR TIK TOK BE DESCRIBED USING THIS METHOD? ”

Ideally, both platforms fall in the class of social networks that can be described using our method as (i)

they are directed networks and (ii) they are based on User-Generated Content. However, we would like to emphasize that our method describes the formation of a sub-network of users with a specific interest in common. Therefore, the model can rather represent the formation of communities within these platforms. Unfortunately, collecting data from Instagram or TikTok is currently against their Terms of Service. On the other hand, also Twitch is a very popular platform among the young generations, and it provides a public API which allows for efficient and extensive data collection. Hence, we opted for Twitch for our case study. It is worth mentioning that, before we collect and analyze the Twitch data set, we did some “pilot data analysis” for another UGC-based online platform called Bilibili, which is very popular in China. On this platform, users can submit their videos labeled by different topics. We manually collect the data of the most popular content contributors on the topics “computer hardware” and “movie summary”. We find that, in either case, the number of followers for the top content contributors also satisfies the Zipf’s law. However, we did not further work on this online platform because collecting data from Bilibili is against its terms of Service.

Based on the above response, in the discussion section of the revised manuscript we explicitly added a call for similar studies on different platforms.

Another direction consists in the analysis of the network spreading characteristics, with particular emphasis on the influencers [27]. Ideally, this should be coupled with empirical analysis on different platforms, e.g., Instagram or TikTok, which are predominant among the new generations.

[R3:6] “ DOES THE "MERITOCRATIC PRINCIPLE" REALLY HELP EXPLAINING THE QUALITY OF UGC? HOW ABOUT THE ROLE OF PROFILING AND ARTIFICIAL CUSTOM AUDIENCE IN THE DECISION MAKING PROCESS OF HUMAN USERS? ”

Thank you for your comment. We agree with the reviewer that recommendation systems implemented in many of today’s online platforms can have a relevant impact on the social network formation process. In particular, it is absolutely interesting to understand to what extent recommendation systems can potentially destroy the results of an intuitive meritocratic principle by providing users with personalized content. We definitely consider this an important research question which we have transparently and explicitly discussed in the conclusions.

While the scope of our paper was primarily the introduction of a previously untouched User-Generated Content quality mechanism in the network formation process, we elaborated some preliminary analysis on the effect of the recommendation systems on our model. The analysis, also described in our response to comment [R1:2], aims at comparing the effect of replacing the previous meeting process with a preferential attachment based rule. Practically, we consider two scenarios: in the first one (already described in the previous version of the manuscript), at each time step, each user meets another distinct user with a probability taken from a uniform distribution. In the second scenario, the probability distribution is built according to the preferential attachment mechanism: the highest in-degree nodes are more likely to be found. This second mechanism should mimic the effect of a recommendation system, which increases the chance of finding other users which are already popular. Despite the different meeting process mechanisms, in both scenarios we leave the individual decision process untouched: users decide to follow the newly met agents if the quality observed is higher than the maximum quality of the currently followed users. The comparison between the two scenarios is presented in our response to comment [R1:2], and it shows that the meritocratic principle still holds: while in the beginning some low-quality users can exceptionally be rewarded by the preferential attachment mechanism, in the long run only the high-quality users will keep on receiving connections (due to the individual decision-making process which is based on a quality threshold). Ultimately, the Zipf’s law of the expected in-degree as a function of the quality ranking is (almost completely) preserved.

Based on the above response, in the revised manuscript, we introduce a section named “Preferential

attachment meeting process” in which we discuss some of the differences between the scenario with the uniform distribution and the one with the preferential attachment in-degree based.

[R3: 7] “ ALTERNATIVELY, DOES YOUR APPROACH SOLVE AN OPEN PROBLEM IN THE FIELD OF MACHINE LEARNED MODELS OF HUMAN BEHAVIOR? ”

Thank you for this comment. The scope of our research is not to solve open problems in the field of machine learned models of human behavior. As a matter of fact, our goal is to build a social network formation model which is sociologically well-funded, and which can describe the process occurring on today’s most popular platforms, which are dominated by the User-Generated Content and offer the possibility of searching users by interests. In order to provide such a model, we first recognize the meritocratic principle in the Twitter data-set of social network scientists. Then, to validate our model, we verify its predictions by comparing its theoretical and numerical results with empirical data collected from Twitch. In essence, we do not construct our model starting from data (as in a typical machine learning approach). Rather, our model is based on the simple and intuitive first principle that online users seek for good-quality contents. We use the empirical data to verify whether our model can describe the underlying network formation process. By doing so, we aim at shading light on the microscopic (individual) linking process that ultimately determines the macroscopic social network structure.

[R3: 8] “ I URGE THE AUTHORS TO SEE MY CRITICISM IN A POSITIVE MANNER I.E. TO ADD TO THE INTRODUCTION MORE MOTIVATION AND TO ADD AT THE END A DISCUSSION OF THEIR RESULTS E.G. BY COMPARING THEM TO REAL DATA (AND NOT A CHERRY-PICKED NETWORK FORMED BY A VANISHING SUBSET OF USERS OF TWITTER OR TWITCH), RATHER THAN SIMULATIONS OR ALTERNATIVELY TO EXPLAIN THE RELEVANCE OF THEIR MODEL AND CORRECTIONS TO AN ACTUAL EXPERIMENT I.E. TO MAKE A FALSIFIABLE PREDICTION ABOUT A YET UNOBSERVED SOCIAL NETWORK (FOR EXAMPLE ABOUT THE RISE OF A YET UNDISCOVERED INFLUENCER AND CHECK IF SHE BECOMES ONE AND IF SO WHEN). ”

Thank you for your comment. According to their suggestion, we revised the introduction to make our motivation and our contribution more clear.

With respect to the real-data analysis, we would like to emphasize that our quality-based model focuses on sets of users which have a certain (strong) common interest, e.g., they are researchers in the field of complex social networks, or they are on-line chess players. As such, the networks we consider for our validation part are not cherry-picked, but they are collected based on this common interest principle, as explained in the manuscript and in Supplementary Note 2. By construction, they constitute sub-networks of the whole Twitter or Twitch networks, and we agree with the reviewer that they are certainly not comparable in size with them. Yet, these interest-based networks still contain on the order of $O(10^6)$ nodes, among which nodes with million(s) of followers can have a significant societal impact on their audience. Based on the above considerations, we believe that the data we have collected and analyzed are appropriate for the validation of our quality-based model which focuses on the formation of communities which revolve around a common interest.

Concerning the last comment, we absolutely agree with the reviewer that predicting the rise of a yet undiscovered influencer is an interesting and very promising research direction. Yet, we believe that accurately testing this type of predictions constitutes by itself an entirely new project, which necessities a sufficiently long time horizon, to make and verify the model’s predictions. Furthermore, we believe that, thanks to the introduction of the User-Generated Content concept, our quality-based model already represents a step forward in the modeling of today’s online platforms. Thanks to its simplicity, the model can be further enriched with important elements, e.g., the recommendation systems, and later properly matched with empirical longitudinal data.

Based on the above response, in the revised manuscript we extensively modified the introduction to make our motivation and contribution more clear. Furthermore, among the future directions in the final discussion, we now include the possibility of an empirical validation on the prediction of the rise of yet undiscovered influencers.

Another direction consists in the analysis of the network spreading characteristics, with particular emphasis on the influencers [27]. Ideally, this should be coupled with empirical analysis on different platforms, e.g., Instagram or Tik Tok, which are predominant among the new generations. In addition, longitudinal data could be used to make interesting predictions on the rise of new influencers.

We finally thank the reviewer again for all their constructive comments which prompted a lot of changes in the paper and led to an improved manuscript.

References

- [1] Albert-László Barabási and Réka Albert. Emergence of scaling in random networks. *science*, 286(5439):509–512, 1999.
- [2] Guido Caldarelli, Andrea Capocci, Paolo De Los Rios, and Miguel A Munoz. Scale-free networks from varying vertex intrinsic fitness. *Physical review letters*, 89(25):258702, 2002.
- [3] Arpita Ghosh and Preston McAfee. Incentivizing high-quality user-generated content. In *Proceedings of the 20th international conference on World wide web*, pages 137–146, 2011.
- [4] Vito DP Servedio, Guido Caldarelli, and Paolo Buttà. Vertex intrinsic fitness: How to produce arbitrary scale-free networks. *Physical Review E*, 70(5):056126, 2004.
- [5] Guido Caldarelli, Andrea Capocci, Paolo Rios, and Miguel A Munoz. Scale-free networks without growth or preferential attachment: Good get richer. *arXiv preprint cond-mat/0207366*, 2002.
- [6] Aaron Clauset, Cosma Rohilla Shalizi, and Mark EJ Newman. Power-law distributions in empirical data. *SIAM review*, 51(4):661–703, 2009.
- [7] Béla Bollobás, Christian Borgs, Jennifer T Chayes, and Oliver Riordan. Directed scale-free graphs. In *SODA*, volume 3, pages 132–139, 2003.
- [8] Mirjam Wattenhofer, Roger Wattenhofer, and Zack Zhu. The youtube social network. In *Sixth international AAAI conference on weblogs and social media*, 2012.
- [9] Shan-Yun Teng, Mi-Yen Yeh, and Kun-Ta Chuang. Toward understanding the mobile social properties: An analysis on instagram photo-sharing network. In *2015 IEEE/ACM International Conference on Advances in Social Networks Analysis and Mining (ASONAM)*, pages 266–269. IEEE, 2015.
- [10] Homa Hosseinmardi, Sabrina Arredondo Mattson, Rahat Ibn Rafiq, Richard Han, Qin Lv, and Shivakant Mishra. Detection of cyberbullying incidents on the instagram social network. *arXiv preprint arXiv:1503.03909*, 2015.
- [11] A Grabowski, N Kruszewska, and RA Kosiński. Properties of on-line social systems. *The European Physical Journal B*, 66(1):107–113, 2008.

- [12] A Grabowski. Human behavior in online social systems. *The European Physical Journal B*, 69(4):605–611, 2009.
- [13] Nan-Nan Li, Ning Zhang, and Tao Zhou. Empirical analysis on temporal statistics of human correspondence patterns. *Physica A: Statistical Mechanics and its Applications*, 387(25):6391–6394, 2008.
- [14] Emilio Ferrara, Roberto Interdonato, and Andrea Tagarelli. Online popularity and topical interests through the lens of instagram. In *Proceedings of the 25th ACM conference on Hypertext and social media*, pages 24–34, 2014.
- [15] Giorgio Fagiolo. Clustering in complex directed networks. *Physical Review E*, 76(2):026107, 2007.
- [16] Andreas M Kaplan and Michael Haenlein. Users of the world, unite! the challenges and opportunities of social media. *Business horizons*, 53(1):59–68, 2010.
- [17] Michael D Conover, Jacob Ratkiewicz, Matthew Francisco, Bruno Gonçalves, Filippo Menczer, and Alessandro Flammini. Political polarization on twitter. In *Fifth international AAAI conference on weblogs and social media*, 2011.
- [18] Venkatramanan S Subrahmanian, Amos Azaria, Skylar Durst, Vadim Kagan, Aram Galstyan, Kristina Lerman, Linhong Zhu, Emilio Ferrara, Alessandro Flammini, and Filippo Menczer. The darpa twitter bot challenge. *Computer*, 49(6):38–46, 2016.
- [19] Boreum Choi and Inseong Lee. Trust in open versus closed social media: The relative influence of user- and marketer-generated content in social network services on customer trust. *Telematics and Informatics*, 34(5):550–559, 2017.
- [20] Paul Gillin. *The new influencers: A marketer's guide to the new social media*. Linden Publishing, 2007.
- [21] S Venus Jin, Aziz Muqaddam, and Ehri Ryu. Instafamous and social media influencer marketing. *Marketing Intelligence & Planning*, 2019.
- [22] Tom AB Snijders. Stochastic actor-oriented models for network change. *Journal of mathematical sociology*, 21(1-2):149–172, 1996.
- [23] Matthew O Jackson. *Social and economic networks*. Princeton university press, 2010.
- [24] Tom AB Snijders. The statistical evaluation of social network dynamics. *Sociological methodology*, 31(1):361–395, 2001.
- [25] Nicolò Pagan and Florian Dörfler. Game theoretical inference of human behavior in social networks. *Nature communications*, 10(1):1–12, 2019.
- [26] Guido Caldarelli. *Large scale structure and dynamics of complex networks: from information technology to finance and natural science*, volume 2. World Scientific, 2007.
- [27] Brooke Erin Duffy. Social media influencers. *The International Encyclopedia of Gender, Media, and Communication*, pages 1–4, 2020.

REVIEWERS' COMMENTS

Reviewer #2 (Remarks to the Author):

I appreciate the authors' efforts to revise the paper.
My concerns have been satisfactory addressed in the revision.